


# Simulated stable water isotopes during the mid-Holocene and pre-industrial using AWI-ESM-2.1-wiso

Xiaoxu Shi[1,2], Alexandre Cauquoin[3], Gerrit Lohmann[1,4], Lukas Jonkers[4], Qiang Wang[1], Hu Yang[1,4], Yuchen Sun[1,5], and Martin Werner[1]

[1]Alfred Wegener Institute, Helmholtz Center for Polar and Marine Research, Bremerhaven, Germany
[2]Southern Marine Science and Engineering Guangdong Laboratory (Zhuhai), Zhuhai, China
[3]Institute of Industrial Science, The University of Tokyo, Kashiwa, Japan
[4]MARUM Center for Marine Environmental Sciences, University of Bremen, Bremen, Germany
[5]Institute of Tibetan Plateau Research, Chinese Academy of Sciences, Beijing, China

**Correspondence:** Martin Werner (martin.werner@awi.de)

**Abstract.** Numerical simulations employing prognostic stable water isotopes can not only facilitate our understanding of hydrological processes and climate change but also allow for a straightforward comparison between isotope signals obtained from models and various archives. In the current work, we describe the performance and explore the potential of a new version of the Earth system model AWI-ESM, labeled AWI-ESM-wiso, in which we incorporated three isotope tracers into all relevant com-

ponents of the water cycle. We present here the results of pre-industrial (PI) and mid-Holocene (MH) simulations. The model well reproduces the observed PI isotope compositions in both precipitation and sea water, and captures their major differences from the MH condition. The simulated relationship between the isotope composition in precipitation ($\delta^{18}O_p$) and surface air temperature is very similar between the PI and MH conditions, and largely consistent with modern observations despite some regional model biases. The ratio of the MH-PI difference in $\delta^{18}O_p$ to the MH-PI difference in surface air temperature is rea-

sonable over Greenland and Antarctica only when summertime air temperature is considered. An amount effect is evident over the North Africa monsoon domain, where a negative correlation between $\delta^{18}O_p$ and the amount of precipitation is simulated. As an example of model applications, we studied the onset date of MH West Africa summer monsoon (WASM) using daily variables. We find that defining the WASM onset based on precipitation alone may yield erroneous results due to the substantial daily variations of precipitation, which may obscure the distinction between pre-monsoon and monsoon seasons. Combining

precipitation and isotope indicators, we suggest in this work a novel method for identifying the commencement of the WASM. Moreover, we do not find obvious difference between the MH and PI in terms of the mean onset date of WASM.





# 1 Introduction

Stable water isotopologues (in this study $H_2^{16}O$, $H_2^{18}O$ and $HD^{16}O$, hereafter called stable water isotopes), are natural tracers
in the water cycle and can serve as valuable indicators of climate change (Dansgaard, 1964). The compositions of stable water
isotopes are commonly expressed as delta values in terms of permil (‰) deviation from a standard, e.g. the Vienna Standard
Mean Ocean Water (V-SMOW). Various syntheses of water isotope measurements have arisen during the last decades, for ex-
ample the Global Network of Isotope in Precipitation (GNIP) (Schotterer et al., 1996; IAEA, 2018) and the global oxygen-18
database of seawater from the Goddard Institute for Space Studies (GISS) (Bigg and Rohling, 2000; Schmidt and Rohling).
These datasets facilitate a greater comprehension of the current hydrological cycle and associated climate processes. In addi-
tion, numerous attempts have been undertaken to interpret past climate changes from a variety of stable water isotope archives,
such as the polar ice cores (e.g., North, 2004; Petit et al., 1999), tropical ice cores (e.g., Thompson et al., 1995, 1994), subtrop-
ical speleothems (e.g., Ku and Li, 1998; Fleitmann et al., 2004) and planktonic/benthic foraminifera (e.g., Shackleton et al.,
1983; McManus et al., 1999). The translation from isotopic records to past climate fluctuations is commonly carried out by
using the observed modern spatial slope between isotope composition and physical variables (e.g., temperature, salinity of the
seawater) as a surrogate for the temporal slope of the past. Typical examples include the reconstruction of Antarctic temper-
ature at glacial–interglacial time scales based on isotope records from Antarctic ice cores (Petit et al., 1999; Augustin et al.,
2004; Jouzel et al., 2007; Kawamura et al., 2007), and reconstruction of the Northern Hemisphere temperature from Greenland
ice core measurements (North, 2004; Thomas et al., 2007; Guillevic et al., 2013; Buizert et al., 2014). Moreover, isotopic ratios
documented in subtropical speleothems can reflect past monsoon rainfall variability with a high temporal resolution (Johnson
et al., 2006; Maher, 2008).

Nevertheless, using isotope data from different archives to draw quantitative inferences about past climate variability re-
mains challenging. For instance, the isotope-based temperature reconstruction over Greenland during the Last Glacial Max-
imum (LGM) is prone to biases due to changes in the seasonality of precipitation (e.g., Werner et al., 2000). Changes in
the primary moisture source areas or air mass transport trajectories during the LGM can have a significant impact on the
isotope-temperature relationship, which has a potential to invalidate the isotopic paleothermometer approach based on the use
of modern observations (Delaygue et al., 2000b; Werner et al., 2001). During the last deglaciation, the temporal slope between
$\delta^{18}O$ and temperature over Greenland is found to be much lower than the spatial slope obtained from modern measurements
(Guillevic et al., 2013; Buizert et al., 2014). Evidence from previous studies has shown heterogeneous isotope-temperature
relationship at East Antarctic plateau during interglacial periods (Sime et al., 2009; Cauquoin et al., 2015). Over subtropical
continents, the isotopic signature in speleothems has often been used as an indicator for the intensity of monsoon rainfall,
however, this isotopic signal is sensitive to a number of other variables, including local temperature, specific humidity, and
atmospheric circulation (Lachniet, 2009). Therefore, it is possible that the isotopic composition determined from subtropical
speleothems does not exclusively represent the local precipitation rate.

Over the past few decades, isotope-enabled models have evolved as valuable, well-established tools, improving our un-
derstanding of the relationship between water isotopes and climate variables. Explicitly, the water isotopes, $H_2^{18}O$ and HDO,





have been incorporated into relevant components of the hydrological cycle in atmospheric general circulation models (GCMs) (Joussaume et al., 1984; Jouzel et al., 1987; Hoffmann et al., 1998; Mathieu et al., 2002; Schmidt et al., 2005; Lee et al., 2007; Risi et al., 2010; Werner et al., 2011; Nusbaumer et al., 2017; Okazaki and Yoshimura, 2019), oceanic GCMs (Schmidt, 1998; Paul et al., 1999; Delaygue et al., 2000a; Wadley et al., 2002; Xu et al., 2012; Liu et al., 2014), land surface models (Riley and Berry, 2002; Fischer, 2006; Yoshimura et al., 2006; Haese et al., 2013), as well as coupled Earth system models (Roche et al., 2004; Schmidt et al., 2007; Tindall et al., 2009; Brennan et al., 2012; Roche and Caley, 2013; Werner et al., 2016; Cauquoin et al., 2019; Brady et al., 2019). They can be used to simulate in parallel the isotopic and climatic signals of the past. More importantly, isotope-enabled models permit a straightforward comparison between isotopic measurements and model simulations and offer the opportunity to comprehend the underlying mechanisms responsible for the variations in isotopic composition documented in various archives (Werner et al., 2000; Risi et al., 2012; Phipps et al., 2013; Bühler et al., 2022).

The mid-Holocene (MH, around 6 ka) is one of the foci of the Paleoclimate Model Intercomparison Project (PMIP) (Kageyama et al., 2018) and has drawn considerable interest from paleoclimate researchers (Yin and Berger, 2015; Brierley et al., 2020; Otto-Bliesner et al., 2021). Proxy-based reconstructions of global temperature has shown a Holocene Thermal Maximum (HTM) possibly centered around the MH (Kaufman et al., 2020). However, a recent study emphasizes a spatial heterogeneity of Holocene temperature trends, contrasting the concept of a globally synchronous MH thermal optimum (Cartapanis et al., 2022). Important external forcings that differ between MH and today include the Earth's orbital parameters and greenhouse gas concentrations (Berger, 1977; Köhler et al., 2017). As a result of the changes in orbital parameters, during the MH, solar insolation was higher in the boreal summer and lower in the boreal winter than it is today, resulting in an enhanced seasonal cycle (Kukla et al., 2002; Shi and Lohmann, 2016; Shi et al., 2020; Zhang et al., 2021). Moreover, the greenhouse gas concentration during the MH was lower than today, which caused a modest global cooling superimposed on the orbital effect (Brierley et al., 2020).

Previous studies identified increased Northern Hemisphere monsoonal precipitation on the basis of model simulations (Jiang et al., 2015; Braconnot et al., 2007; Nikolova et al., 2012; Fischer and Jungclaus, 2010) as well as various proxy records (Wang et al., 2008b; Bartlein et al., 2011). Due to the intensification in rainfall, the isotopic composition in precipitation decreases over North Africa and South Asia (Cauquoin et al., 2019). Over Greenland, both model simulation and ice core measurements show an enriched $\delta^{18}$O in precipitation associated with an increase in annual mean temperature during the MH (Cauquoin et al., 2019; North, 2004). Most climate models participating in the PMIP4 simulate an enhanced Northern Hemisphere summer monsoon during the MH periods relative to the present-day (D'Agostino et al., 2019). Further evidence of this is also documented in pollen-based proxy records (Bartlein et al., 2011). However, there is a lack of knowledge on the initiation date of MH summer monsoon due to low temporal resolution in proxy data. Since models can give adequate high-frequency variables to explore synoptic phenomena, we can study the possible characteristics of monsoon onset using the model outputs with high temporal resolution under both PI and MH climatic conditions.

In the present study, we put our focus on 3 major aspects: First, we have developed a new Earth system model with enabled stable water diagnostics, called hereafter AWI-ESM-wiso. This model inherits the atmosphere and land surface components from MPI-ESM-wiso (Cauquoin et al., 2019). We additionally implemented 3 isotopic tracers (i.e., $H_2^{16}O$, $H_2^{18}O$, and HDO)





into the ice-ocean module FESOM2 (Danilov et al., 2017), all of which are treated as passive tracers in the ocean, in the same way as in the oceanic component of MPI-ESM-wiso (i.e., MPIOM). Compared to MPIOM, FESOM2 has the advantage of adopting a much higher spatial resolution in the region of interest (Fig. S1). Second, we provide the preliminary results of AWI-ESM-wiso simulations under both pre-industrial (PI) and MH boundary conditions, with an emphasis on the global distribution of isotope compositions and the relationship between water isotopes and the climatic variables during both time periods. We aim to give insight into the MH with the use of a state-of-the-art high-resolution Earth system model with the capability to simulate isotopic compositions in all relevant hydrological components. Finally, with the utility of model outputs in daily frequency, we analyze the initiation date of West Africa summer monsoon onset in the PI and MH eras.

## 2 Methodology

### 2.1 Model components description

Stable water isotopes have been incorporated into all relevant components of the hydrological cycle in AWI-ESM, a state-of-the-art coupled climate model developed at the Alfred Wegener Institute (AWI), which is an extension of the AWI climate model version 2 (AWI-CM2) (Sidorenko et al., 2019). The atmospheric component of the model is ECHAM6 (Stevens et al., 2013) which also contains a land-surface module (JSBACH) representing multiple plant functional types and two types of bare surface (Loveland et al., 2000; Raddatz et al., 2007). Parallel to the water cycle, 3 isotope tracers (i.e., $H_2^{16}O$, $H_2^{18}O$, and HDO) have been implemented in ECHAM6 (Cauquoin et al., 2019; Cauquoin and Werner, 2021), in a similar way as done by earlier studies (Jouzel et al., 1987; Hoffmann et al., 1998; Werner and Heimann, 2002; Werner et al., 2011). These tracers are handled as separate forms of water, which are described in an identical manner to bulk moisture in all aggregate states (gaseous, liquid, and solid water) when no phase transition occurs. Differences between isotope tracers and bulk moisture arise only during processes of phase transformation, where the ratios of fractionation is calculated based on different vapor pressures and diffusivities between the isotope tracers. For more details about the isotope implementation, we refer to previous studies (Werner et al., 2011; Cauquoin et al., 2019; Cauquoin and Werner, 2021).

The ice-ocean component FESOM2 has been built on the basis of the previous version FESOM1.4, (Danilov et al., 2004; Wang et al., 2008a), but with improved numerical efficiency. The model utilizes a multi-resolution dynamical core and is based on triangle grid and finite volume discretization (Danilov et al., 2017). For the present study, we added three isotopic variables in FESOM2, representing $H_2^{16}O$, $H_2^{18}O$, and HDO. They are treated as passive tracers and are freely advected and diffused within the ocean. Equilibrium fractionation takes place during formation/growth of sea ice. For this process, the corresponding equilibrium fractionation coefficients suggested by Lehmann and Siegenthaler (1991) are used. During the melting of sea ice and the snow lying on the sea ice, it is explicitly assumed that no isotopic fractionation occurs. Therefore the meltwater contains the same isotopic ratio as its source, either the sea ice itself or the snow lying on top of it. During ocean-atmosphere interactions, isotopic compositions of surface water are modified by a number of fractionation processes (e.g., evaporation of surface seawater) and non-fractionation processes (e.g., rainfall flux into the ocean, river runoff).





AWI-ESM-wiso employs the OASIS3-MCT coupler (Valcke, 2013) with an intermediate regular exchange grid. Mapping
between the intermediate grid and the atmospheric/oceanic grid is handled with bilinear interpolation. The atmosphere compo-
nent calculates 12 climatic air–sea fluxes based on four surface climatic fields provided by the ocean module. To enable stable
isotope diagnostics in the model, we add additional 6 atmospheric isotopic variables transported to the ocean, including the
mass fluxes of $H_2^{16}O$, $H_2^{18}O$, and HDO in both liquid and solid water forms, which then modify the isotope compositions of the
sea surface water. The ocean module calculates the isotopic ratio for all isotope tracers at the ocean surface with respect to the
V-SMOW standard, and transfers the values to the atmosphere. The model has been widely used with its standard configuration
for modern climate condition (Sidorenko et al., 2019), the mid-Holocene (Shi et al., 2022a), the last interglacial (Kageyama
et al., 2021b; Otto-Bliesner et al., 2021), the last glacial maximum (Kageyama et al., 2021a) as well as a transient simulation
covering the last 6,000 years (Shi et al., 2022a).

## 2.2   Experimental design

With AWI-ESM-wiso, we first perform a time-slice simulation for the pre-industrial (PI) period, i.e. the year 1850 CE. We
initiate the PI atmosphere with the mean climatology obtained from an Atmospheric Model Intercomparison Project (AMIP)
simulation (Roeckner et al., 2004), which has been performed with prescribed sea surface temperatures (SST) and sea ice
concentration (SIC) from 1978 to 1999. The initial condition of the ocean is based on the climatological temperature and
salinity from the World Ocean Atlas (WOA) for the period 1950-2000 (Levitus et al., 2010). The initial isotope composition
in the atmosphere is defined as $\delta^{18}O$=-20 ‰; $\delta D$=-150 ‰ according to the V-SMOW scale, and the water isotope ratios in
the ocean are initialized with constant zero values, i.e., $\delta^{18}O$=0 ‰; $\delta D$=0 ‰. The PI simulation has been integrated for 1,500
model years under the boundary condition suggested by the most recent phase of PMIP (Otto-Bliesner et al., 2017). Branched
from our PI experiment, we conduct a MH run with a total length of 1,500 model years. According to the protocol of PMIP
(Otto-Bliesner et al., 2017), orbital parameters are computed following Berger (1977), and the concentrations of greenhouse
gas are derived from the polar ice cores (Flückiger et al., 2002; Monnin et al., 2004; Schneider et al., 2013; Schilt et al.,
2010; Buiron et al., 2011). Specific values for the boundary conditions are shown in Table 1. In our simulations, the dynamic
vegetation is interactively calculated via the land surface model JSBACH. Both experiments are configured on T63L47 grid for
the atmosphere, i.e., 47 levels in the vertical direction and a mean horizontal resolution of about 1.9x1.9 degree that corresponds
to a horizontal grid spacing of approximately 140x210 km at mid-latitudes. The ocean component applies a spatially-variable
mesh as described in Koldunov et al. (2019) and Scholz et al. (2022) , with a resolution of about 100 km in the open ocean, 25
km over polar areas and 35 km along coastlines. A refined grid with a resolution up to 35 km is employed for the equatorial belt.
We detect only minor trends throughout the final 100 model years of both simulations (Table 1), indicating that the simulated
climate and isotopes are in a state of quasi-equilibrium. In order to illustrate the climatology pattern of each variable, we use
mean values over the last 100 model years.





**Table 1.** Upper part: boundary conditions for pre-industrial and mid-Holocene experiments. Lower part: trends in global mean surface temperature (GMST) and mean oceanic temperature, salinity and $\delta^{18}$O at 3000 m depth of the global ocean.

| Experiment | PI | MH |
|---|---|---|
| $CO_2$ (ppm) | 284.3 | 264.4 |
| $CH_4$ (ppb) | 808.2 | 597 |
| $N_2O$ (ppb) | 273 | 262 |
| Eccentricity | 0.016764 | 0.018682 |
| Obliquity | 23.459° | 24.105° |
| perihelion - 180° | 100.33° | 0.87° |
| GMST (K/century) | 0.001 | 0.003 |
| 3000 m temperature (K/century) | 0.002 | -0.005 |
| 3000 m salinity (psu/century) | -0.00008 | -0.0006 |
| 3000 m $\delta^{18}$O (‰/century) | -0.006 | -0.005 |

## 2.3 Data

To evaluate how well our model represents modern isotopic patterns, we compare our PI results with a compilation of isotope in precipitation data and an $\delta^{18}$O of seawater assimilation product based on both models simulations and observations. Performing a pre-industrial simulation instead of a present-day one, which is a slightly warmer climate, probably adds a small negative bias in our modeled temperatures, and therefore in the modeled isotopic composition, compared to these data.

### 2.3.1 GNIP database

GNIP is a worldwide network for monitoring $\delta^{18}$O and $\delta$D in precipitation (IAEA, 2018), whose measurements were initiated in 1960 mainly by the International Atomic Energy Agency (IAEA) and the World Meteorological Organization (WMO). As in previous studies (Werner et al., 2016; Cauquoin et al., 2019), in our work only the records from stations in operation for at least five years within the period 1961 to 2007 are used. This subset of data is used to validate our modeled isotopic content of precipitation under PI conditions.

### 2.3.2 Ice core data

Like in Cauquoin et al. (2019), isotope measurements from a subset of 10 Antarctic (WAIS Divide Project Members, 2013) and 5 Greenland ice cores (Sundqvist et al., 2014) are selected and compared with our model results for the PI. If data for the MH is available as well, we also compare the MH-minus-PI isotope composition between model simulations and ice core data. For each individual ice core record, we take the averaged $\delta^{18}$O value over the last 200 years and over the time interval of 6 $\pm$ 0.5 ka as a representative mean PI and MH value, respectively.



### 2.3.3 Speleothem calcite data

The Speleothem Isotope Synthesis and Analysis (SISAL) dataset provides a global compilation of speleothem $\delta^{18}O$ records from 455 speleothems from 211 cave sites spanning the period from the Last Glacial Maximum (LGM) until the present (Comas-Bru et al., 2020). The $\delta^{18}O$ in speleothem calcites ($\delta^{18}O_c$) is expressed with respect to the Pee Dee Belemnite (PDB) standard. Therefore the transformation equations described in (Coplen et al., 1983) are applied to convert the speleothem $\delta^{18}O$ value from the PDB to V-SMOW scale, to enable a straightforward comparison between model results and speleothem records. Caution should be taken during model-data comparison as the speleothem $\delta^{18}O$ signals might be associated with seasonal climate conditions, for example speleothems from western Germany are more sensitive to the winter temperature and precipitation due to a strong influence of winter rainfall on the oxygen isotope composition of cave drip water (Wackerbarth et al., 2010). Following Comas-Bru et al. (2019), in the present study we select 30 speleothem sites (33 cores) for which isotopic values are available for both the MH (defined as the time interval of $6 \pm 0.5$ ka) and PI periods (1850–1990 CE).

### 2.3.4 Marine calcite data

Jonkers et al. (2020) have recently compiled a multi-parameter marine data synthesis that contains time series spanning 0 to 130 ka. It is a data product focusing exclusively on time series with a robust chronology based on benthic foraminifera $\delta^{18}O$ and radiocarbon dating. The data set contains totally 896 time series of eight paleoclimate variables from 143 individual sites, among which 174 of planktonic foraminifera $\delta^{18}O$ and 205 samples of benthic foraminifera $\delta^{18}O$ are available. For both planktic and benthic foraminifera, a diversity of species have been measured. For more detailed information of this data set, we refer to Jonkers et al. (2020).

To enable a direct comparison between model results and isotopic signals in the calcite shells of foraminifera, the equation described in Shackleton (1974) which links temperature during calcite formation (T) to the equilibrium fractionation of inorganic calcite precipitation around 16.9 $^o$C are used :

$$T = 16.9 - 4.38 \times (\delta^{18}O_{c(PDB)} - \delta^{18}O_{oce(PDB)}) + 0.1 \times (\delta^{18}O_{c(PDB)} - \delta^{18}O_{oce(PDB)})^2 \tag{1}$$

In equation (1), $\delta^{18}O_{c(PDB)}$ represents the isotopic composition of calcite on the Pee Dee Belemnite (PDB) scale, which can be converted to SMOW scale via $\delta^{18}O_{c(SMOW)} = \delta^{18}O_{c(PDB)} + 0.27$ (Hut, 1987).

### 2.3.5 Assimilation product

A dynamically consistent database of global three-dimensional $\delta^{18}O$ of seawater has been produced by Breitkreuz et al. (2018). This set of data was generated based on an optimized simulation using an oceanic GCM constrained with climatological salinity and temperature data observed during 1951-1980 and global $\delta^{18}O$ collected from 1950 to 2011. An adjoint method for variational data assimilation was used for optimization, enabling a consistency between the simulation and the observation. One thing to be noted is that the dataset shows a certain degree of bias in the surface levels in the Arctic Ocean, which is caused





by the rather low resolution applied in the model (2.8°x2.8°) as well as a lack of effect from isotopically highly depleted precipitation on the ocean in areas covered by sea ice. The final data was interpolated and provided with a regular 1°x1° grid.

## 3 PI validation for water isotopes

The distribution of simulated precipitation weighted annual mean $\delta^{18}O$ in precipitation (hereafter referred to as $\delta^{18}O_p$) is depicted in Fig. 1a. The most notable characteristic is the decrease in $\delta^{18}O_p$ with increasing latitude. This is due to the favorable correlation between surface temperature, condensation temperature and the isotopic composition of meteoric water, particularly at high latitudes (Dansgaard, 1964). The so-called continental effect is also evident in Fig. 1a especially over Eurasia, where $\delta^{18}O_p$ gets lighter toward the continental interior as the air masses moving inland experience multiple cycles

of condensation and precipitation. Moreover, $\delta^{18}O_p$ is typically lighter at higher altitudes such as the Himalayas and the Alps due to the so-called altitude effect.

To evaluate the performance of AWI-ESM-wiso in simulating the $\delta^{18}O_p$ distribution under PI climate conditions, we compare measurements from the GNIP database, ice cores, and 33 selected speleothem records with simulated values at the same locations. Following the methodology described in the preceding section, we have used 70 values from the GNIP database

which reflect the mean $\delta^{18}O_p$ throughout the course of at least five calendar years. The results are shown in the circles of Fig. 1a and in the scatter plot of Fig. 1b, in which the reference 1:1 line represents a perfect model-data match. As illustrated in Fig. 1b, our model agrees well with the observed $\delta^{18}O_p$ based on various archives with a root mean square error (RMSE) of 2.245 ‰. It is a big challenge for isotope-enabled models to reproduce correctly the isotope composition at sites with very low temperatures. A typical example is Antarctica, where earlier model simulations tend to overestimate the values of $\delta^{18}O_p$ due to

a possible warming bias. Compared to earlier model studies (e.g., Werner et al., 2016; Cauquoin et al., 2019), AWI-ESM-wiso significantly reduces this model-data discrepancy over the Antarctic continent.

We compare our modeled annual mean $\delta^{18}O_{oce}$ in sea surface water with the assimilation-based data set of Breitkreuz et al. (2018). As seen from Fig. 2a, the most enriched $\delta^{18}O_{oce}$ as simulated by AWI-ESM-wiso can be found in the Mediterranean and Red Sea (more than 1.5 ‰) and secondly in North Atlantic Ocean with $\delta^{18}O_{oce}$ values reaching 1.2 ‰, due to intense

evaporation and net freshwater export. In general, $\delta^{18}O_{oce}$ decreases from mid to high latitudes as a consequence of the latitudinal temperature gradient. Since the subtropics experience more evaporation than the tropics, the isotopic composition of the subtropic surface waters is higher than that of the equatorial regions. Furthermore, the most depleted $\delta^{18}O_{oce}$ can be found in the Arctic Ocean, Baffin Bay, and, to a lesser degree, the Pacific Ocean north of 40°N. Similar patterns can be seen in the assimilated data as in Fig. 2b.

Despite the model-data agreement on the aforementioned significant aspects in spatial structures of sea surface $\delta^{18}O_{oce}$, there are a number of discrepancies identified between model and data in terms of the $\delta^{18}O_{oce}$ magnitudes. For instance, our model underestimates the isotopic composition of the South Atlantic and overestimates the $\delta^{18}O$ values in the Southern Ocean.

For further evaluation, the simulated and assimilated zonal mean $\delta^{18}O_{oce}$ at depths for both the Atlantic and Pacific sectors are shown in Fig. 2c-f. In the Atlantic, the most enriched $\delta^{18}O_{oce}$ values are found near the ocean's surface and subsurface





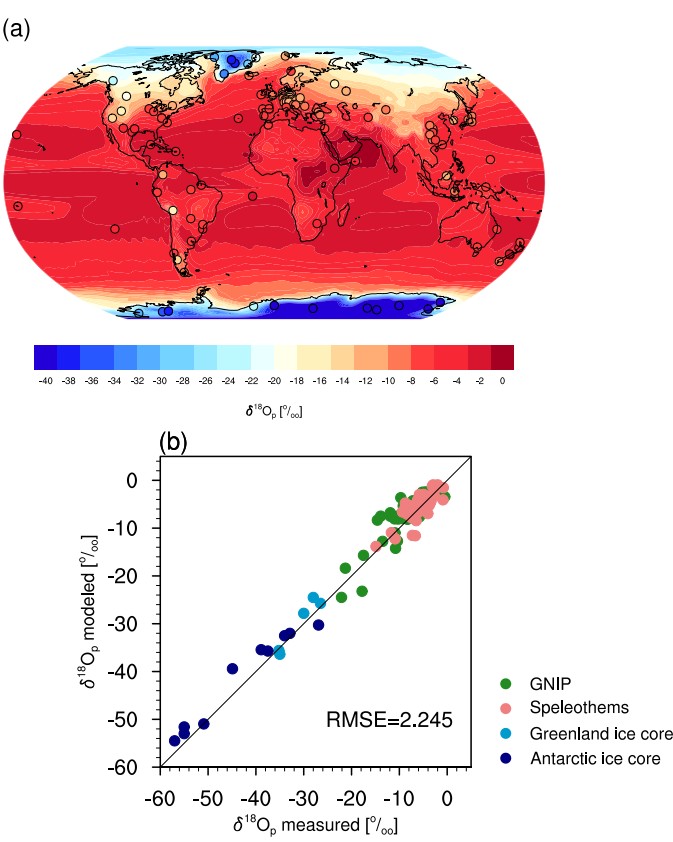

**Figure 1.** (a) Simulated pre-industrial (shading) and observed present-day (circles) precipitation-weighted annual mean $\delta^{18}O$ in precipitation. (b) Scatter plot of modeled-versus-measured $\delta^{18}O_p$. Observation values are based on GNIP database (IAEA, 2018), speleothems (Comas-Bru et al., 2020), and ice core records (WAIS Divide Project Members, 2013; Sundqvist et al., 2014).



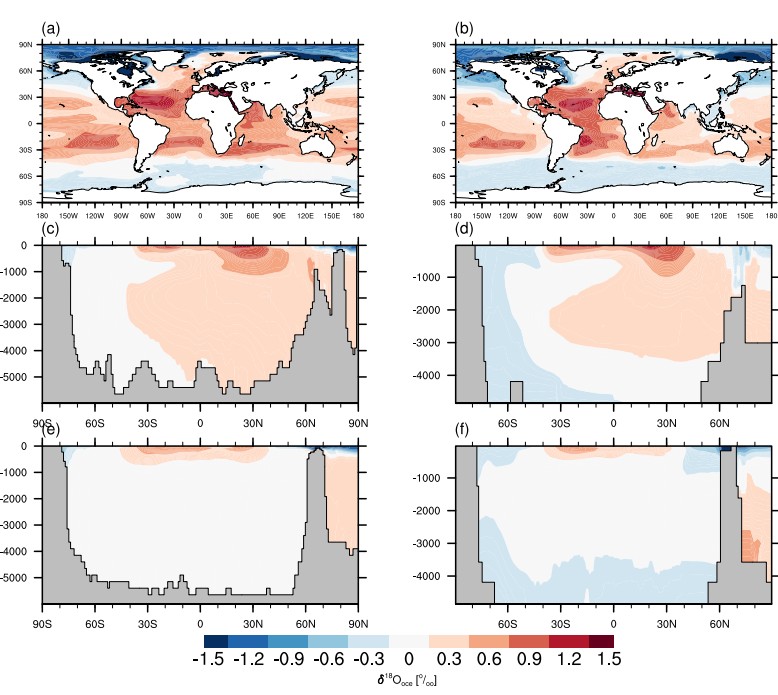

**Figure 2.** (a,b) Modeled (a) and assimilated (b) annual mean $\delta^{18}\mathrm{O}_{oce}$ in surface ocean from 0 to 10 m. (c,d) Modeled (c) and assimilated (d) zonal mean $\delta^{18}\mathrm{O}_{oce}$ across the Atlantic section. (e,f) As in (c,d) but for the Pacific section.





due to considerable influence of air-sea interaction. The North Atlantic Deep Water (NADW) is isotopically enriched, although

with a smaller magnitude in $\delta^{18}O_{oce}$ than the upper layers. Deep water convection in the Atlantic Ocean is responsible for

the enrichment in NADW $\delta^{18}O_{oce}$. Both the Antarctic Intermediate Water (AAIW) and Antarctic Bottom Water (AABW)

are depleted as they are supplied by Southern Ocean surface water, which has negative $\delta^{18}O_{oce}$ values. The distributions of

$\delta^{18}O_{oce}$ in the subsurface and deep waters of the Pacific are distinctly different from that of the Atlantic. Since the deep water

formation of the Pacific is substantially weaker than the Atlantic, the enriched seawater is confined to depths of 0–1000 m in

the tropics and subtropics. Pacific intermediate and bottom depths are primarily dominated by negative $\delta^{18}O_{oce}$.

Our simulated zonal mean $\delta^{18}O_{oce}$ are in good agreement with Breitkreuz et al. (2018), for both the Atlantic and Pacific

basins. However, in certain regions, our simulation deviates from the data in the magnitudes of $\delta^{18}O_{oce}$. An example is the

Southern Ocean, where our model exhibits enrichment biases across the whole water column. Moreover, AWI-ESM-wiso tends

to overestimate the $\delta^{18}O_{oce}$ values of Antarctic and Pacific bottom water.

These comparisons demonstrate that AWI-ESM-wiso is capable of capturing the present-day spatial distribution of $\delta^{18}O$ in

precipitation and seawater, as indicated by the consistency between model results and available data sets. It strengthens our

confidence in the quality of the MH simulation examined in the subsequent sections.

## 4    Climate and isotope differences between MH and PI

### 4.1    Differences in climate signals

Stable water isotopes are closely linked to changes in standard climate variables, among which the most important ones are the

surface air temperature (SAT), precipitation, and evaporation. Therefore before examining the isotope anomalies between MH

and PI, we firstly examine the simulated climate differences.

Fig. 3 depicts the simulated anomalies in seasonal and annual mean SAT, which imply an enhanced seasonality in the

Northern Hemisphere, i.e., a cooling of boreal winter and a warming of boreal summer, consistent with results form model

ensemble means (Brierley et al., 2020). Such seasonality enhancement is caused by changes in the orbital parameters which

lead to redistribution of latitudinal and seasonal incoming solar insolation at the top of the atmosphere. Specifically, more

insolation is received in JJA than in DJF during MH relative to PI. Another prominent aspect is the year-round cooling of the

Sahara, in conjunction with a northward migration of the tropical rain-belt, as well as increased precipitation and cloudiness

over Sahara. Such cooling coincides with other climate models (Brierley et al., 2020), but contradicts proxy records which

document a general warming over that region (Bartlein et al., 2011). The warming registered in pollen-based proxy, however,

mainly reflects earlier growth season during the MH (Bartlein et al., 2011). During DJF, Antarctica presents a moderate cooling,

while in JJA it is characterized by non-uniform changes with a warming in the east and cooling in the west.

Simulated changes in annual mean precipitation between MH and PI are shown in Fig. 4a. The most striking pattern is

the enhanced precipitation over the North Africa monsoon region, which is associated with a northward displacement of the

Intertropical Convergence Zone (ITCZ). There is also an increase in precipitation, albeit to a lesser extent, in other Northern

Hemisphere monsoon regions such as North America and South Asia. The tropical Pacific experiences a substantial drying in





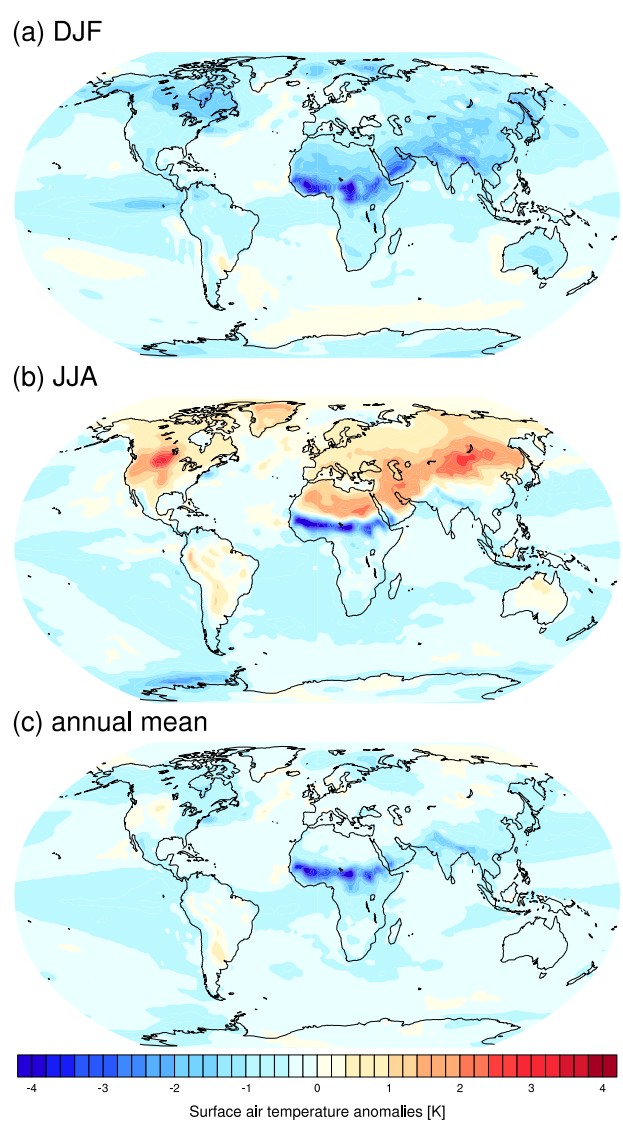

**Figure 3.** Simulated anomalies (MH-PI) of surface air temperature for (a) DJF, (b) JJA, and (c) annual mean. Units: K.





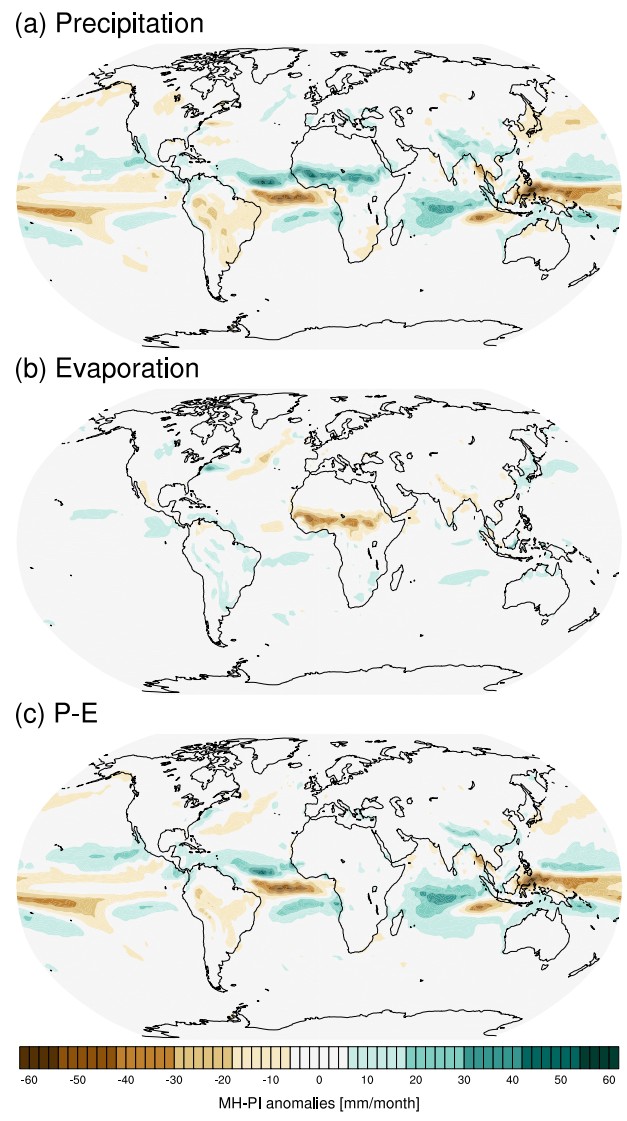

**Figure 4.** Simulated anomalies (MH-PI) of annual mean (a) precipitation, (b) evaporation (positive downward), and (c) precipitation minus evaporation. Units: mm/month.





MH compared to PI, while the Indian Ocean experiences wetter conditions. In general, our results regarding the MH-minus-PI precipitation agree well with other climate models (Brierley et al., 2020) as well as pollen-based reconstructions (Bartlein et al., 2011). The evaporation anomalies over the globe are relatively small, except for the North Africa where large anomalous evaporation occurs (Fig. 4b). The distribution of changes in net precipitation, defined as precipitation minus evaporation (P-E), as shown in Fig. 4c, resembles that of the precipitation changes, with an exception over North Africa where no clear change in P-E is detected due to a compensation between the enhancement in both precipitation and evaporation.

### 4.2 Differences in isotopic signals

In this section we turn to analyze the anomalous isotope composition between MH and PI. The shadings in Fig. 5a represents the simulated changes in the precipitation-weighted annual mean $\delta^{18}O_p$. Although only modest annual mean warming (less than 0.2 K) is found over Greenland and the Arctic, this nevertheless leaves a noticeable imprint on the isotope composition of precipitation, with $\delta^{18}O_p$ enriched by up to 1 ‰ in MH with respect to PI. Such change may mainly reflect the regional warming during summer. Another clear picture from Fig. 5a is the depletion of $\delta^{18}O_p$ over North Africa (more than -4 ‰) and South Asia (up to -2 ‰), closely correlated with increased monsoonal rainfall and reduced surface air temperature. The Antarctic continent is generally dominated by enriched $\delta^{18}O_p$ except for localized regions that are slightly depleted. Over the remaining land surfaces, AWI-ESM-wiso simulates small to moderate negative MH-PI anomalies down to -1 ‰. Concerning the ocean region, our model presents more positive $\delta^{18}O_p$ values over the Indo-Pacific warm pool, as a consequence of reduced precipitation during the MH. In addition, large positive $\delta^{18}O_p$ anomalies in the Amundsen Sea are produced by our model because of a pronounced JJA warming in MH relative to PI.

We assess our simulated MH-minus-PI $\delta^{18}O_p$ against ice core and speleothem calcite data, which are shown in Fig. 5a (circles) and Fig. 5b. Our model results generally agree with the proxy records for the $\delta^{18}O_p$ changes over Greenland, Europe, and part of North America. However, while AWI-ESM-wiso simulates a moderate change in $\delta^{18}O_p$ across the globe (i.e., -1.3 ‰ to +0.4 ‰), the speleothem data reveals a wide range of $\delta^{18}O_p$ changes, being from -3.4 ‰ to +2 ‰ (Fig. 5b). Spatially, such bias can detected over the South Asia, tropical Africa, South America and western Antarctica, where our model severely underestimate the change in $\delta^{18}O_p$.

Fig. 6a shows the simulated changes of annual mean $\delta^{18}O_c$ in ocean surface water between MH and PI versus available reconstruction data based on planktonic foraminifera (Jonkers et al., 2020). We only choose species which cover more than 6 locations on a global scale. Here $\delta^{18}O_c$ refers to $\delta^{18}O$ in calcite, calculated from the simulated $\delta^{18}O$ in seawater and ocean temperature upon Shackleton (1974). This conversion is required to enable a direct comparison between model results and isotopic signals in the calcite shells of planktonic foraminifera (see Methodology Section). It is apparent from Fig. 6a that in our model the Pacific Ocean, South Atlantic, and Southern Ocean are more $\delta^{18}O_c$-enriched during MH compared to PI, generally in line with the proxy records with the exception of a few proxy reconstructions exhibiting a depletion signal. A depletion in the $\delta^{18}O_c$ across the Indian Ocean is simulated in our model, as a result of an increase in net precipitation. The scatter plot of Fig. 6b indicates that our simulated annual mean $\delta^{18}O_c$ anomalies fall within a narrower range (-0.2 ‰ to 0.2 ‰) compared to the reconstructed anomalies (ranging from -0.5 ‰ to 0.5 ‰). The calculated RMSE between modeled and measured MH-PI $\delta^{18}O_c$





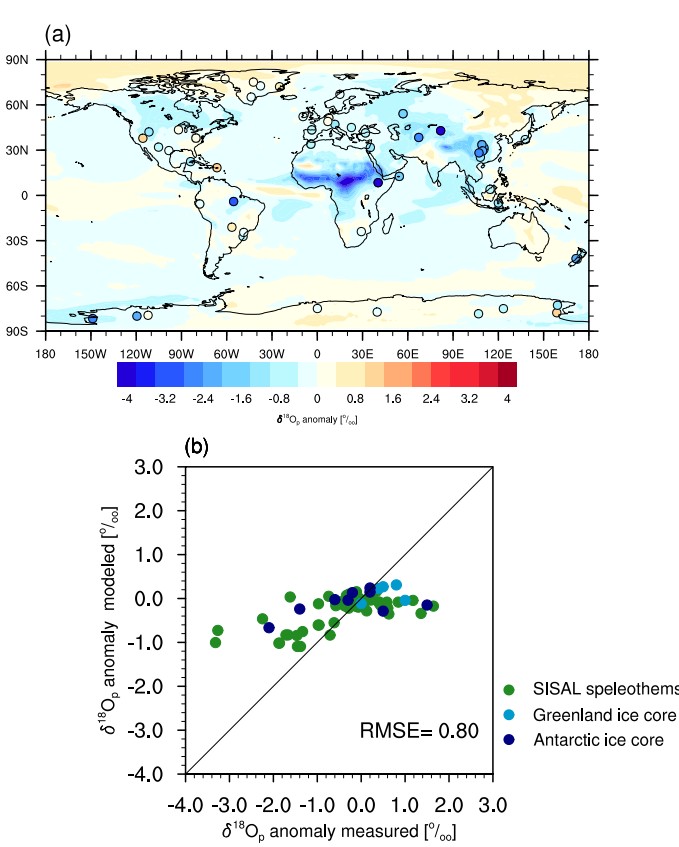

**Figure 5.** (a) Simulated (shading) and reconstructed (circles) anomalies in precipitation-weighted annual mean $\delta^{18}$O in precipitation between MH and PI. (b) Scatter plot of modeled-versus-reconstructed $\delta^{18}$O$_p$ anomalies. Units: ‰.

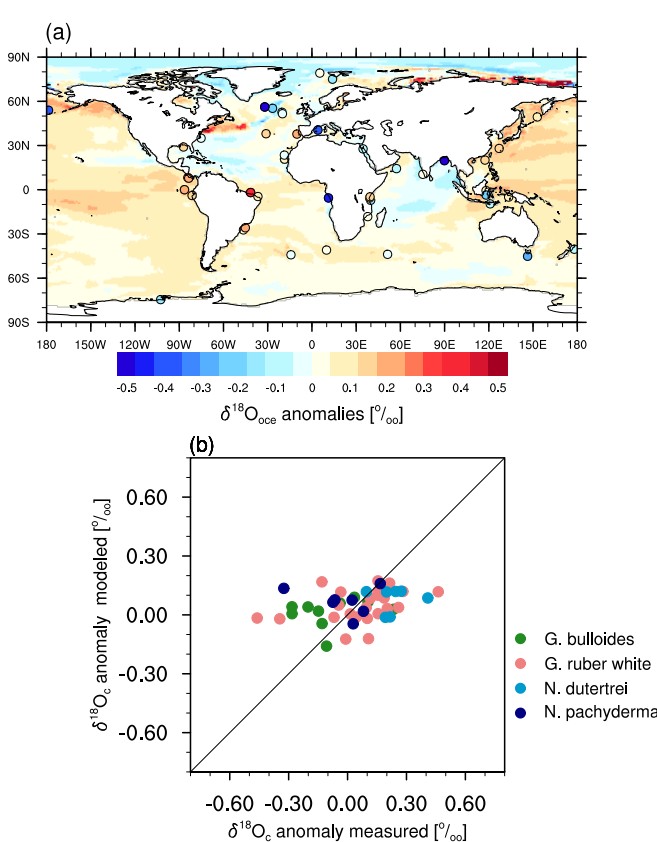

**Figure 6.** (a) Simulated (shading) and reconstructed (circles) anomalies in annual mean $\delta^{18}O_c$ in ocean surface waters between MH and PI. (b) Scatter plot of modeled-versus-reconstructed $\delta^{18}O_c$ anomalies. Units: ‰.





changes is 0.18 ‰ for the species of *Globigerina bulloides*; *Globigerinoides ruber albus*; *Neogloboquadrina dutertrei*, and *Neogloboquadrina pachyderma*. The modeled $\delta^{18}O_c$ anomalies are mostly close to the *Neogloboquadrina pachyderma*-based measurements, nonetheless, our model result deviates largely from the observation for one *Neogloboquadrina pachyderma*

location documenting a significant depletion in $\delta^{18}O_c$, leading a RMSE of 0.38 ‰ for this planktonic foraminifera species.

## 5   Climate-isotope relationship in MH and PI

A strong positive correlation between the annual mean surface temperature and the precipitation-weighted $\delta^{18}O_p$ was firstly recognized by Dansgaard (1964). This relationship has served as the foundation for numerous paleoclimate research in which the temporal evolution of past temperature for a certain region are derived from isotopic proxies. The observed present-day

spatial slope between isotope and temperature is widely used as a surrogate for the temporal gradient at reconstruction sites. Nevertheless, the temporal relationship between isotopes and temperature may vary over time. Moreover, the temporal isotope-temperature slope may differ from the observed spatial slope at present-day. Therefore, it is essential to examine: 1. Is the modeled spatial and temporal isotope-temperature relationship under PI and MH conditions identical to the observed modern spatial gradient at various regions? 2. Under which condition can we support the application of modern spatial isotope-temperature

gradient in reconstructing past climate changes? To answer these questions, in the following we perform further analysis on the spatial and temporal relationship between $\delta^{18}O_p$ and temperature based on both PI and MH simulations. In addition, the temporal relationships between isotope and precipitation are investigated as well.

### 5.1   Spatial $\delta^{18}O_p$-temperature gradient

To determine the simulated global spatial $\delta^{18}O_p$-temperature slope, we use the $\delta^{18}O_p$ and surface air temperature modeled

at all grid cells with an annual mean temperature below 20 $^{\circ}C$. This criterion is taken to restrict our analysis to regions with a dominant temperature dependency. Our modeled global spatial $\delta^{18}O_p$-temperature slope has a value of 0.71±0.005 ‰/$^{\circ}C$ for PI (Fig. 7a, here the uncertainty is calculated from the interannual standard deviation), similar to the value of 0.69 ‰/$^{\circ}C$ derived from observations (Dansgaard, 1964). Compared to former studies using isotope-enabled climate models (Cauquoin et al., 2019; Risi et al., 2010; Gierz et al., 2017), our result is better in line with observation. The same as PI, our MH experiment

also yields a global spatial $\delta^{18}O_p$-temperature slope of 0.71±0.005 ‰/$^{\circ}C$ (Fig. 7f). To compute the isotope-temperature slope of polar regions, we take only grid boxes around the ice core locations. Over Greenland, our modeled $\delta^{18}O_p$-temperature gradients under present-day and MH conditions are 0.76±0.042 and 0.74±0.045 ‰/$^{\circ}C$, respectively (Fig. 7b,g), higher than the value obtained from modern observations (0.67 ‰/$^{\circ}C$) (Johnsen et al., 1989) and previous model studies using MPI-ESM-wiso (0.71 ‰/$^{\circ}C$) (Cauquoin et al., 2019) and ECHAM4 (0.58 ‰/$^{\circ}C$) (Werner et al., 2000). For Antarctica, our model

generates a spatial isotope-temperature slope of 0.76±0.016 ‰/$^{\circ}C$ for PI and 0.75±0.018 ‰/$^{\circ}C$ for MH (Fig. 7c), slightly lower than the mean observed value of 0.8 ‰/$^{\circ}C$ (Masson-Delmotte et al., 2008). For East Antarctica under both present-day and mid-Holocene conditions, our simulated spatial $\delta^{18}O_p$-temperature gradient is around 0.75 ‰/$^{\circ}C$. In the PI experiment, the gradient for West Antarctica, is a little bit larger (0.79±0.026 ‰/$^{\circ}C$) than for the East Antarctic. But for MH we find



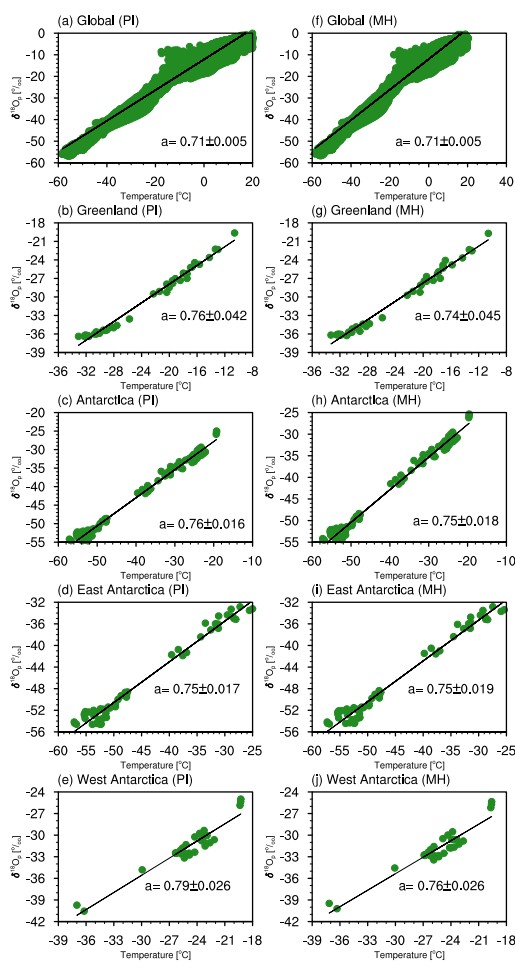

**Figure 7.** Simulated PI spatial $\delta^{18}O_p$-temperature gradient for (a) all global grid boxes with a annual mean temperature below 20 $^\circ C$, (b) Greenland, (c) Antarctica, (d) East Antarctica, and (e) West Antarctica. (f-j) Same as in (a-e), but for the MH.

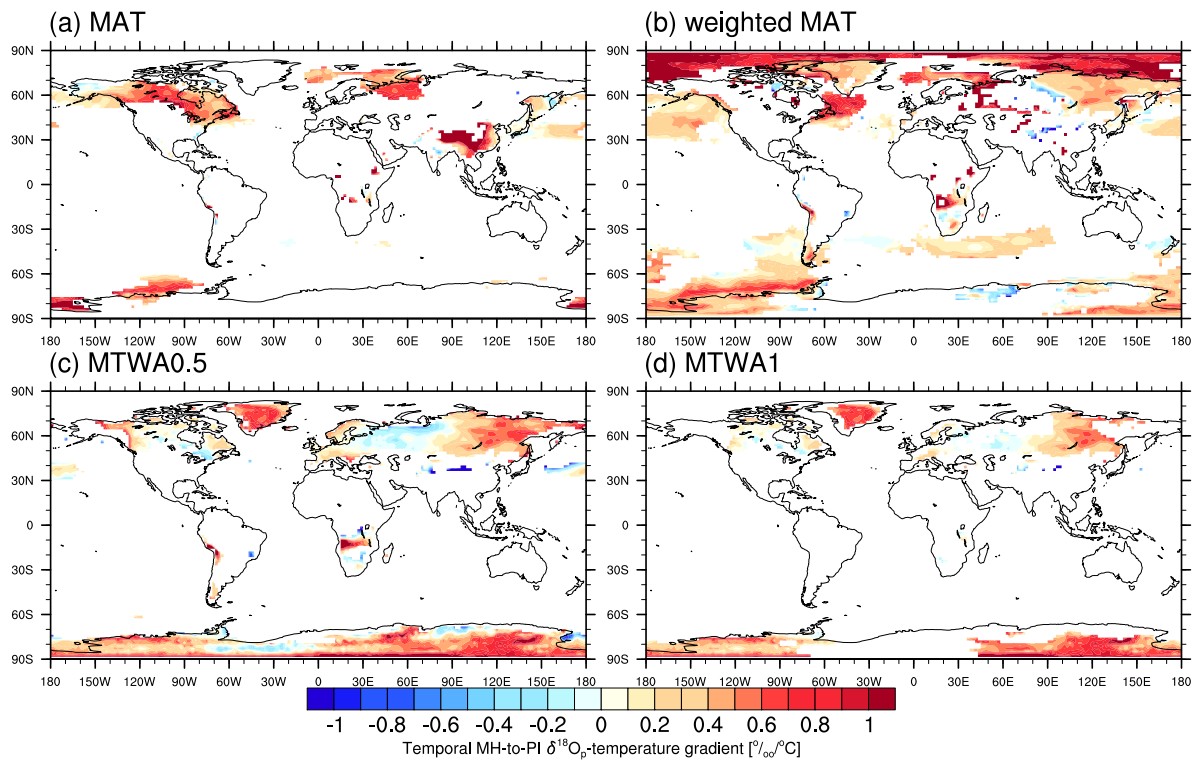

**Figure 8.** (a) Simulated temporal MH-to-PI $\delta^{18}O_p$-temperature gradient for all grid boxes with mean annual temperature (MAT) for both PI and MH being lower than 20 $^\circ C$ and the absolute change in temperature between MH and PI being at least 0.5 $^\circ C$. (b) Same as in (a), but using the precipitation weighted annual mean temperature. (c) Same as in (a), but using the temperature of the warmest month (MTWA). (d) Same as in (c) but with the absolute change of MTWA being at least 1 $^\circ C$.Units: $\%o/^\circ C$.

no clear distinction of the spatial isotope-temperature relationship between West and East Antarctica. Our simulation results

suggest similar $\delta^{18}O_p$-temperature relationship between MH and PI, both globally and regionally. This also indicates that the moderate changes in $\delta^{18}O_p$ and $\delta^{18}O_{oce}$, simulated in our model, are more likely related to weak responses of the climate to MH boundary conditions.





## 5.2  Temporal relationship

Here we examine the simulated MH-to-PI temporal relationship between $\delta^{18}O_p$ and surface air temperature changes, defined

as:

$$m = (\delta^{18}O_{p,MH} - \delta^{18}O_{p,PI})/(T_{MH} - T_{PI}) \tag{2}$$

In accordance with previous studies (Cauquoin et al., 2019; Gierz et al., 2017), we only consider grid points with a mean temperature below 20 $^\circ C$ for both the MH and PI periods, thereby excluding from our analysis tropical regions where local variances of $\delta^{18}O_p$ are strongly controlled by changes in precipitation amount and moisture source region. To avoid numerical

errors in calculated temporal relationships caused by very small MH-PI temperature changes, another criterion is adopted that the absolute change of T between the two time intervals (i.e., $T_{MH} - T_{PI}$) must be non-negligible, namely not less than 0.5 $^\circ C$. Several approaches are available for T in equation (1). As a first step we define T as the mean annual temperature (MAT) and we show in Fig. 8a the modeled MH-to-PI temporal gradient between $\delta^{18}O_p$ and MAT changes. Over southeast of China, the gradients approach 0.9 ‰/$^\circ C$, significantly greater than the observed global modern spatial gradient of 0.69 ‰/$^\circ C$ (Dansgaard,

1964). This high value obtained from our model is attributable to the combined impacts of an increase in monsoonal rainfall (Fig. 4a) and a decrease in surface air temperature (Fig. 3). Greenland and the majority of Antarctica, where the polar ice core samples are collected, are dominated by missing data as the MAT changes there are not substantial (Fig. 3).

In the part that follows, T is defined as the simulated mean temperature of the warmest month (MTWA), i.e., the mean temperature of July and January for the Northern and Southern Hemispheres, respectively, and the calculated global distribution

of temporal gradients is given in Fig. 8c. As seen, $\delta^{18}O_p$-MTWA temporal gradients are present across mid and high latitudes of the Northern Hemisphere, with the bulk of values varying from 0.2 to 0.8 ‰/$^\circ C$. Over Greenland, the mean temporal gradient averaged over ice core sites is 0.64 ‰/$^\circ C$, close to the observed modern spatial value of 0.67 ‰/$^\circ C$. The East and West Antarctic respectively presents a mean coefficient of 0.55 ‰/$^\circ C$ and 0.39 ‰/$^\circ C$. However, certain locations of the Antarctica are found to have negative $\delta^{18}O_p$-MTWA temporal gradients. This reflects either a "selection" or a "recorder" problem (Gierz

et al., 2017). The former may arise when temperature is not the primary factor affecting the isotope signal, implying that the $\delta^{18}O_p$ changes are strongly influenced by other processes such as the precipitation amount, moisture origin, and water transport pathways. In this instance, a more strict threshold for the temperature change might be beneficial. The "recorder" problem is associated with pronounced change of seasonality or intermittency of the precipitation rate (Sime et al., 2009). This problem can be reduced by the use of precipitation weighted mean annual temperature instead of arithmetic MAT.

To decrease the "selection" problem, we increase the threshold for MH-minus-PI temperature to 1 $^\circ C$, and the resulting global map of temporal $\delta^{18}O_p$-MTWA gradients is presented in Fig. 8d. Applying this constraint on MTWA removes the most negative gradients from the Antarctic continent. The mean temporal gradient becomes 0.60 ‰/$^\circ C$ for both Greenland and Antarctica ice core locations. Our results suggest that the spatial $\delta^{18}O_p$-T slope observed under modern climate could be a surrogate for the MH-to-PI temporal isotope-temperature gradient during the warmest month/season over Greenland and

Antarctica.



**Table 2.** Mean temporal MH-to-PI $\delta^{18}O_p$-T gradient averaged over the ice core locations for different T definitions, as well as the observed spatial $\delta^{18}O_p$-T gradient at present-day. Units: ‰/°C

| T definition | Greenland | Antarctica | East Antarctica | West Antarctica |
|---|---|---|---|---|
| MAT | - | - | - | - |
| MTWA0.5 | 0.64 | 0.48 | 0.55 | 0.39 |
| MTWA1 | 0.60 | 0.60 | 0.68 | 0.51 |
| weighted MAT | 0.36 | 0.22 | 0.15 | 0.29 |
| Observation | 0.67 | 0.79 | 0.85 | 0.84 |

In addition, to address the "recorder" problem, we recalculated the temporal $\delta^{18}O_p$-T gradients with T being the precipitation weighted mean annual temperature, in the same way as in Gierz et al. (2017) and Cauquoin et al. (2019). We obtain a mean temporal $\delta^{18}O_p$-T gradient of 0.36 ‰/°C for Greenland and 0.22 ‰/°C for the Antarctica, much lower than the observed spatial slopes at present-day (Table 2). Apparent from Fig. 8b, there are still a number of grid cells with a negative gradient,

indicating that the change of seasonality or intermittency of the precipitation is not the primary reason for the negative gradient values.

To calculate the MH-to-PI temporal annual mean $\delta^{18}O_p$-precipitation gradient, we only consider the grid boxes with a substantial precipitation anomaly (larger than 0.5 mm/day) to ensure a dominate precipitation dependency. Fig. 9a presents the distribution map of the temporal gradient between $\delta^{18}O_p$ and total precipitation. The most remarkable feature occurs over the

North Africa monsoon domain, where our model simulates a strong negative $\delta^{18}O_p$-precipitation gradient up to -4 ‰ mm$^{-1}$d, associated with the increased rainfall rate as shown in Fig. 4a. Our findings are consistent with the so-called amount effect. For other regions, no obvious MH-PI $\delta^{18}O_p$-precipitation gradient is found. If summer mean $\delta^{18}O_p$ and precipitation are considered instead of the annual mean values, the gradient pattern remains the same (Fig. 9b). This is in contrast to the results of MPI-ESM-wiso which indicate a much steeper gradient for the yearly mean than for the JJA-mean values (Cauquoin et al.,

380    2019).

## 6   Onset date of the West Africa summer monsoon

It is widely believed that the Northern Hemisphere summer monsoon is enhanced in the MH as compared to present-day. So far it is unknown whether the wetness in MH is due to increased precipitation alone or combined with an extension in the monsoon length. Though the monsoon onset has been explored in a great number of studies based on modern observations,

there is a lack of knowledge on the initiation date of MH summer monsoon due to low temporal resolution in proxy data. In this section for the first time we examine the onset of mid-Holocene West Africa summer monsoon (WASM) using the simulated monsoon-related changes of both climatic and isotopic variables (i.e., precipitation, $\delta^{18}O_p$, and dex$_p$) on a daily scale.

Precipitation on a daily scale is often used for calculating the monsoon onset (e.g., Pausata et al., 2016; Sultan and Janicot, 2003). According to Pausata et al. (2016), the WASM onset is defined as the date preceding the largest increase in precipitation





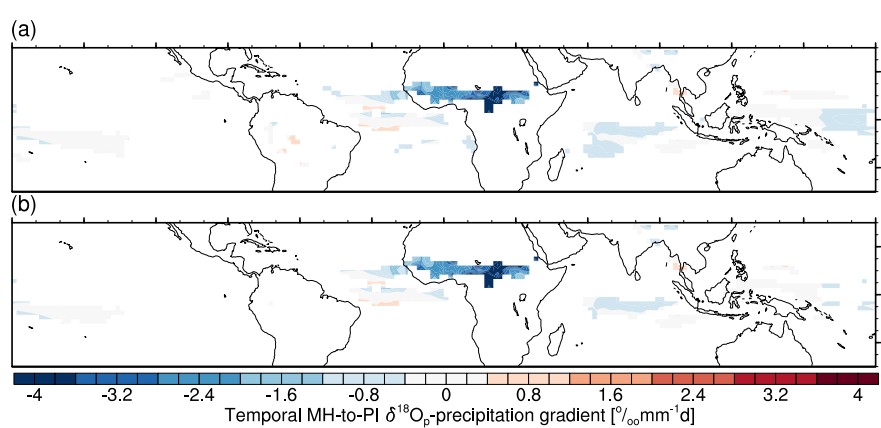

**Figure 9.** Simulated temporal MH-to-PI $\delta^{18}O_p$-precipitation gradient for (a) annual mean and (b) JJA-mean values. Units: ‰mm$^{-1}$d.



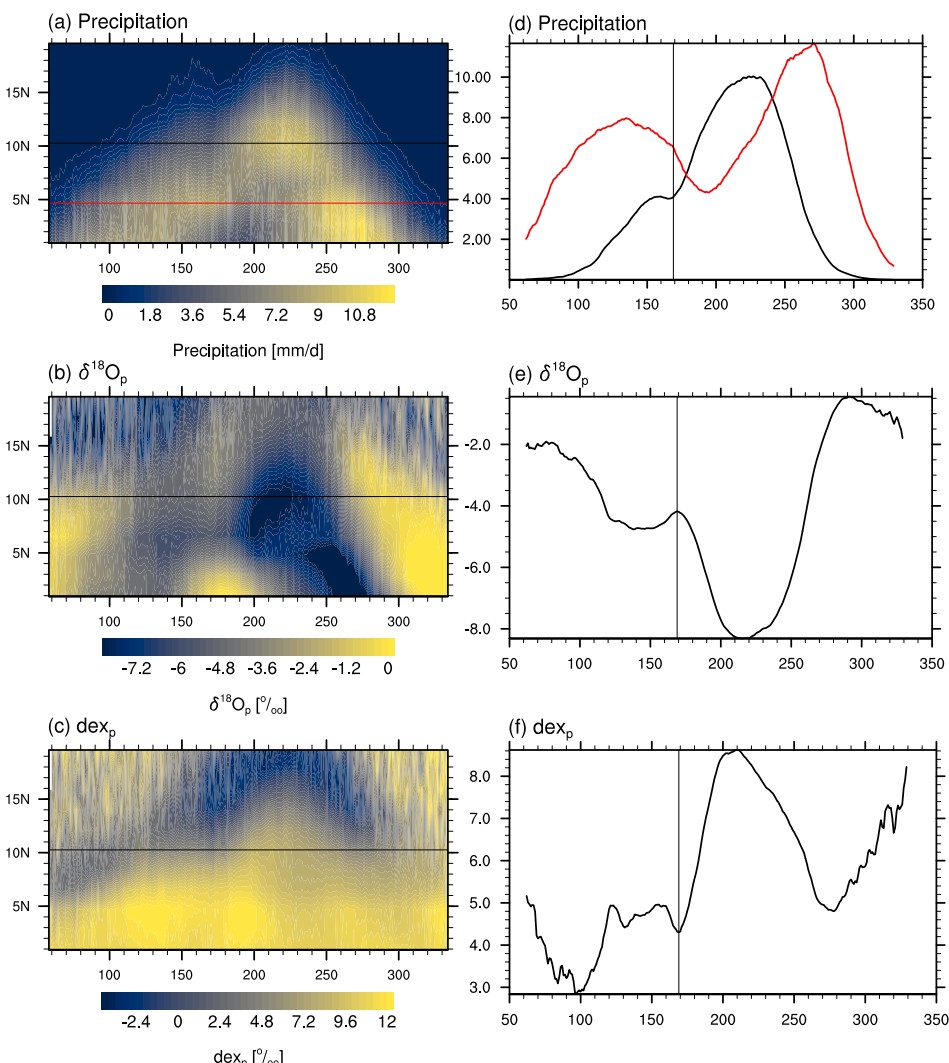

**Figure 10.** Simulated temporal time–latitude diagram of climatological daily (a) precipitation, (b) $\delta^{18}O_p$, and (c) $dex_p$ averaged over 15°W - 20°E under MH climate condition, with x-axis representing the day of the year and y-axis the latitude. The two black lines respond to 4.7°N and 10.3°N. (d) Time sections of precipitation averaged over 15°W - 20°E at 10.3°N (black line) and at 4.7°N (red line), filtered with high frequency variability (period less than 10 days) removed. (e,f) Time sections of (e) $\delta^{18}O_p$ and (f) $dex_p$ averaged over 15°W - 20°E at 12.3°N, filtered with high frequency variability (period less than 10 days) removed.





over a 20-day period at the latitude of the maximum in the first Empirical Orthogonal Function (EOF) of the zonally averaged rainfall over the West Africa monsoon domain (15°W-20°E, 0-20°N). Following Pausata et al. (2016), we first perform an EOF analysis on the MH zonally averaged daily rainfall data from March to November. The first two EOFs explain 79.7% and 12.3% of the precipitation variability. As a second step, the latitudinal locations of the maxima of the first two EOFs (10.3°N, and 4.7°N) are computed, representing the ITCZ position during monsoon and pre-onset time respectively. The WASM onset

occurs in conjunction with a rapid northward shift of the ITCZ. Fig. 10a shows a time–latitude diagram of climatological daily precipitation averaged over the West Africa monsoon domain (15°W-20°E, 0-20°N) under MH climate condition. The beginning of WASM leads to a fast increase in precipitation at the north location of ITCZ (10.3°N) as well as a decrease in precipitation at the south location (4.7°N), a sign of a northward shift of the ITCZ. The monsoon onset is defined as the date preceding the largest increase in precipitation at the north location over a 20-day period.

Besides daily precipitation, isotope signals also offer an alternative approach to examine the onset of summer monsoon. The work of Risi et al. (2008) indicates the possibility of detecting the onset of West Africa summer monsoon with the use of the isotope composition in precipitation, as the monsoon onset is accompanied by a rapid decrease in $\delta^{18}O_p$ due to intense rainfall as well as an increase in $dex_p$ due to reduced re-evaporation of falling rain drops in a humid environment. This phenomenon is also evident in our model. As seen in Fig. 10e,f, the commencement of WASM is marked by a peak in $\delta^{18}O_p$ and a trough

in $dex_p$. To calculate the monsoon onset, we define $I=\delta^{18}O_p - dex_p$, then, a peak in $I$ is identifiable when WASM begins (a typical example shown in Fig. S2). Determining the day when $I$ reaches its peak value from June to July is therefore a straightforward method for determining the onset of WASM. However, in certain years there are multiple $I$ peaks in June and July (as illustrated in Fig. S3). In order to lessen the overall degree of unpredictability, precipitation can be used as a secondary control. Using combined isotope and precipitation indicators, we propose the following strategy for identifying the WASM

onset:

(1) Calculate the time series of $I$ and precipitation at 12.3°N average over 15°W – 20°E between 1th June and 31 July.

(2) Detect the peaks in $I$.

(3) For each day with a $I$ peak, calculate the trend in precipitation over the following 20 days.

(4) The WASM onset is then defined as the day with the greatest precipitation trend obtained from the previous step.

Due to its continuous and cumulative nature, the isotope composition is a more accurate indicator of the onset of the monsoon, in contrast to precipitation which is typically an individual convection event. As indicated by Risi et al. (2008), the ability of isotope composition in precipitation to document the intra-seasonal regional signal of convective variability is even better than that of the raw local outgoing long wave. Moreover, in our model study, there is a large daily variance in the precipitation over West Africa, leading to occasional difficulty in distinguishing the monsoon season from the pre-onset phase. Two

instances are presented here. In the first case (Fig. S4), the precipitation-based method suggests that the WASM commences on the 207th day of the year, whereas the ITCZ shift occurs between the 150th and 200th day of the year as evidenced by the decrease in precipitation at 4.7°N (red line in Fig. S4c). An additional example is illustrated in Fig. S5, wherein the precipitation rate experiences a gradual increases over the course of 50-210th days of the year, lacking any abrupt changes. In this





instance, identifying the onset of the WASM using solely daily precipitation data is challenging. Therefore, the utilization of

isotope tracers can enhance the calculation of onset date of summer monsoon in West Africa.

Analyses of our PI experiment suggest a mean WASM onset date of June 23th based on the isotope approach with a standard deviation of 15 days, the MH simulation yields a similar result (June 21th ± 11 days), indicating that there is no discernible difference between the MH and PI in terms of the simulated initiation date of WASM. This result further indicates that the simulated intensification in MH WASM is more likely attributable to a rise in precipitation rate than an extension of the

monsoon season. The initiation date of WASM, as determined by precipitation alone, is June 25th ± 14 day for PI, similar to the isotope-based approach. For MH, however, the precipitation-based method yields a much later WASM onset date with a larger variability (i.e., July 1st ± 19 days) than the isotope-based approach. The decline in precipitation at 4.7°N is an important mark of the ITCZ shift associated with the beginning of summer monsoon. Compared to the isotope-based method, the precipitation-based method shows a less pronounced decrease in the composite rainfall time series at 4.7°N around the

onset date (Fig. 11c) for MH. This is due to the fact that in certain model years the dry-wet season transition over North Africa is not very clear, or the increase in monsoonal rainfall is more pronounced during the mid-or-late stage of monsoon season than in the onset stage, therefore the precipitation-based method may identify a much later WASM onset than the true date. In addition, our histogram plot (i.e., Fig. 12) shows that the WASM onset defined by the isotope indicator is within 150- 200th day of the year, in contrast, there are 22% model years in the MH where the precipitation-based method determines a WASM onset

later than July 19th (190th day of the year). One guess is that in these model years, the precipitation-based method captures the mature stage of WASM rather than the onset date. In order to test our speculation, In Fig. 13 we re-depict the composite time-latitude diagram of daily precipitation only for the model years in MH with an onset date later than July 19th as calculated by the precipitation-based approach, and the result clearly shows an increasing trend in the averaged rainfall at 4.7°N around the onset date (Fig. 13c), which is unlikely to occur during the northward transition of the ITCZ.

Our investigation on the WASM onset can be treated as a case study of model data application, which may help shed light on how we can use model variables with daily frequency to examine synoptic climatic processes. On the basis of model simulations with stable isotope diagnostics, we show here the potential of enhancing the precipitation-identified technique with the additional use of isotope composition to detect the arrival of past summer monsoon. However, as proxy-based reconstruction cannot resolve daily variations in precipitation or water isotopes, such analysis can only be achieved with the use of daily model

variables.

## 7   Discussion

In this study we have evaluated the performance of AWI-ESM-wiso in simulating the pre-industrial isotope characteristics by a comparison to modern measurements and an assimilation product by Breitkreuz et al. (2018).

Our modeled PI $\delta^{18}O_p$ agrees well with the observation data, not only for (sub)tropical and mid-latitude areas, but even

for the polar regions where temperatures fall below -20 $°C$. These regions have been a major challenge for isotope-enabled models in terms of reproducing correctly the isotope composition in precipitation and surface snow. Using a coarse-resolution



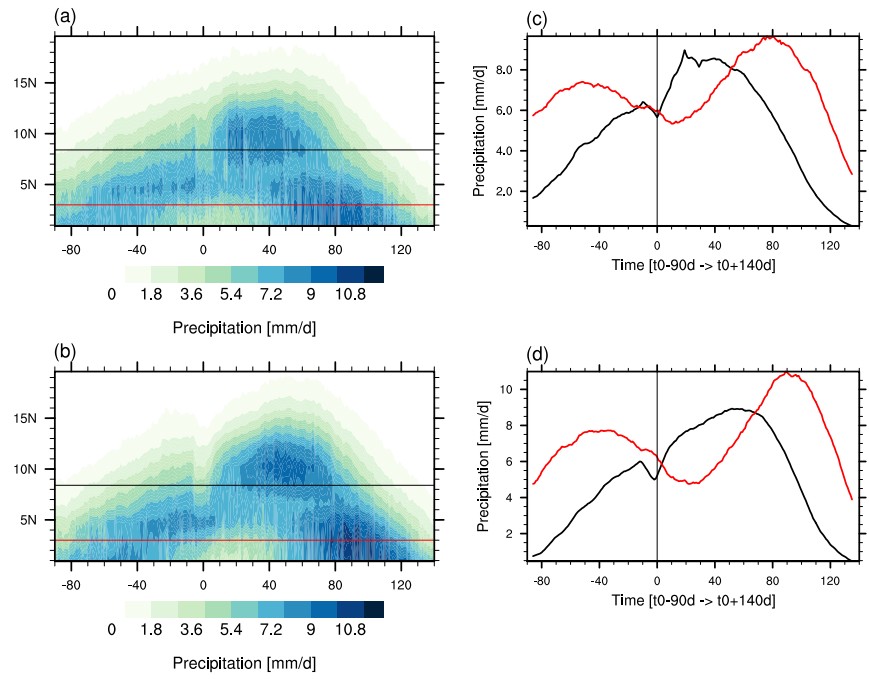

**Figure 11.** (a,b) Simulated composite time–latitude diagram of daily precipitation averaged over 15°W-20°E under MH climate conditions, with x-axis representing the time shift from the onset date (t0), (a) is for the precipitation-based approach, and (b) for the isotope approach. The two black lines respond to 4.7°N and 12.3°N. (c) Time series of diagram (a) at 4.7°N (red line) and 12.3°N (black line). (d) Time series of diagram (b) at 4.7°N (red line) and 12.3°N (black line), filtered with high frequency variability (period less than 10 days) removed. The reference line in (c,d) represents the onset date.

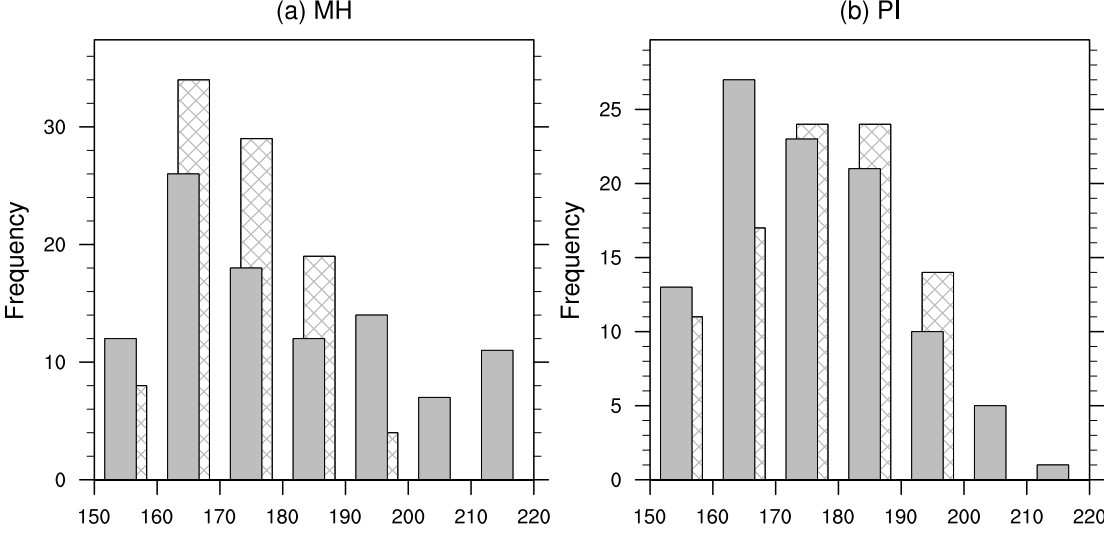

**Figure 12.** Histogram of WASM onset based on the precipitation approach (solid bars) and isotope approach (stippling bars) for (a) MH and (b) PI.

atmosphere model, Werner et al. (2011) obtained an enrichment bias in $\delta^{18}O_p$ across Antarctic ice core locations, mostly linked to an overestimation of Antarctic temperatures. This bias did not improve by coupling the model to the ocean model MPIOM (Werner et al., 2016). The use of a coarse model resolution is a possible reason for this model-data mismatch. An

upgraded model version with higher spatial and vertical resolution for the atmosphere decreased the bias in simulated $\delta^{18}O_p$ over Antarctica, but the modeled values still deviated clearly from observations (Cauquoin et al., 2019). With the use of AWI-ESM-wiso in our study, we obtain a good model-data agreement over Antarctica, even for the ice core locations with temperatures lower than -50 $^{\circ}C$. This is a significant improvement compared to results with earlier models equipped with water isotopes.

Compared to an assimilation dataset (Breitkreuz et al., 2018), our model produces more depleted oxygen-18 in the Arctic surface sea water for PI. However, we should keep in mind that the assimilated $\delta^{18}O_{oce}$ at the surface level of the Arctic Ocean

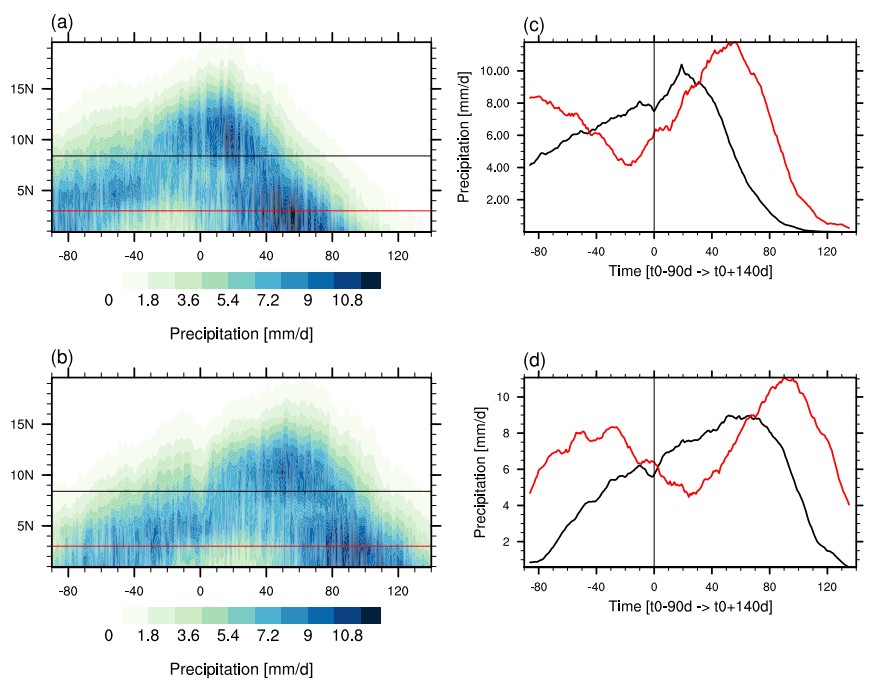

**Figure 13.** Same as in Fig. 11, but for the model years with an onset date being later than July 10th as calculated by the precipitation-based approach.





may be biased because the influence of isotopically highly depleted precipitation is not preserved in the sea ice model used for the assimilation, as pointed out by Breitkreuz et al. (2018). The enrichment bias of $\delta^{18}O_{oce}$ in the Antarctic bottom water in our PI simulation results from an overestimation of $\delta^{18}O_{oce}$ in the Southern Ocean surface water. However, our PI simulation

is constrained by the boundary conditions of 1850 CE, whereas the data set used for comparison was generated based on ocean general circulation model constrained by $\delta^{18}O_{oce}$ data collected from 1950 to 2011 and climatological salinity and temperature data collected from 1951 to 1980. Due to the rise of Greenhouse gases after 1850, enhanced melting of ice shelves may have occurred (Rintoul, 2007), which could result in an isotopically more depleted Southern Ocean.

For the mid-Holocene, AWI-ESM-wiso simulates enhanced seasonality in surface air temperature and increased precip-

itation across the monsoon areas of the Northern Hemisphere, as compared to the pre-industrial period. For assessing the modeled MH-minus-PI isotope anomalies, reconstructions derived from ice cores, speleothems and multiple species of plank- tonic foraminifera are used. Our model captures the signs in the reconstructed MH-minus-PI $\delta^{18}O$ in precipitation and sea water well, however with a much narrower range. Upon comparing model and data, we also discover inconsistencies in certain locations. The mismatch between simulated and observed isotope signals might be attributable to either model bias or data

uncertainties. Compare to pollen-based reconstructions, our modeled MH-PI anomalies in temperature and precipitation are less pronounced (Bartlein et al., 2011; Shi et al., 2022b), which tends to leave a weak imprint on the simulated isotope changes. Moreover, our MH simulation differs from PI only for the insolation and greenhouse forcings, but in reality other elements (e.g., lakes, vegetation, volcanoes, ozone, or the ecosystem) also play important roles. Model resolution is another key factor affecting the simulated result (Werner et al., 2011). On the other hands, there are also uncertainties in the reconstructed data. For

instance, ice core records are also influenced by the seasonality or intermittency of precipitation (Werner et al., 2000). $\delta^{18}O_p$ signals documented in speleothems may result from a complicated interplay of localized environmental processes (Lachniet, 2009). In addition, foraminifera in cold areas are more likely to record summer $\delta^{18}O_{oce}$ values (e.g., Jonkers and Kučera, 2015)

Our analysis of the MH-to-PI temporal gradient between $\delta^{18}O_p$ and temperature indicates that the MH-to-PI temporal ratio of $\delta^{18}O_p$ to surface air temperature is reasonable over Greenland and Antarctica only when summertime air temperature is

considered. Our results suggest that the spatial $\delta^{18}O_p$-T slope observed under modern climate could be a surrogate for the MH-to-PI temporal isotope-temperature gradient during the warmest month/season over Greenland and Antarctica, consistent with a previous study using MPI-ESM-wiso (Cauquoin et al., 2019). Nevertheless, this conclusion might be model-dependent, since both MPI-ESM-wiso and AWI-ESM-wiso use the same atmospheric model, ECHAM6-wiso. Therefore, further analyses of the different slopes by using other isotope-enabled models would be advantageous.

The mid-Holocene Northern Hemisphere summer monsoon has been explored in a great number of studies. It is widely believed that the Northern Hemisphere monsoonal rainfall was stronger during the MH than PI (Jiang et al., 2015; Fischer and Jungclaus, 2010; Bartlein et al., 2011). The subtropic rainfall was isotopically more depleted (Herold and Lohmann, 2009; Cauquoin et al., 2019; Gierz et al., 2017) as a result of enhanced precipitation and stronger advection of moisture from the source ocean surface (Herold and Lohmann, 2009). A variety of studies have attempted to derive the modern summer monsoon

onset based on observations for the region of Asia (Nguyen-Le et al., 2014; Moron and Robertson, 2014; Joseph et al., 2006), Africa (Sultan and Janicot, 2003; Fitzpatrick et al., 2015; Dunning et al., 2016), and North America (Bombardi et al., 2020).





However, because proxy time series cannot resolve daily variations in precipitation or water isotopes, analysis of the onset date of the palaeomonsoon is only possible using climate models. Since the onset of summer monsoon is accompanied by a rapid increase in rainfall, the daily precipitation is often used to define the monsoon onset (Sultan and Janicot, 2003; Dunning et al.,
2016). Other studies have used the surface outgoing longwave radiation (Fontaine et al., 2008; Chenoli et al., 2018; Bhatla et al., 2016) and the low-level zonal wind (Wang et al., 2004; Zhang, 2010) for this purpose. In addition to climatic variables, isotope signals offer an alternative approach to examine the onset of Northern Hemisphere summer monsoon. The validity of $\delta^{18}O_p$ to study Indian monsoon onsets has been supported by Yang et al. (2012). Earlier studies of the $\delta^{18}O_p$ over South Asia have shown a coincidence of dramatic decrease in $\delta^{18}O_p$ with monsoon onset (Tian et al., 2001; Vuille et al., 2005). Moreover,
Risi et al. (2008) have revealed that, during the Africa monsoon onset, the abrupt increase of convective activity over the Sahel is associated with an abrupt decrease in $\delta^{18}O_p$ as well as an increase in deuterium excess in precipitation. Our results are in line with these findings and suggest that combining signals from isotope and precipitation could enhance the definition of WASM onset.

## 8  Conclusions

In the present study we report the first results of two equilibrium simulations performed under both PI and MH boundary conditions using a new developed coupled climate model equipped with water stable isotopes, named AWI-ESM-wiso. For PI, we provide a model evaluation for the simulated isotope composition in precipitation and ocean surface water versus various sources of observations. Our modeled $\delta^{18}O_p$ values are in good agreement with GNIP and ice core measurements. The global distribution of $\delta^{18}O_{oce}$ of the surface water and the zonal mean $\delta^{18}O_{oce}$ in Atlantic and Pacific sections produced by our model
show great similarity with an assimilation product by Breitkreuz et al. (2018).

Our model results in terms of the climate changes between MH and PI are in line with PMIP4 models (Brierley et al., 2020), showing an enhanced seasonality in surface temperature which is driven by the redistribution of seasonal insolation and an increase in Northern Hemisphere monsoonal rainfall governed by a northward shift of the Intertropical Convergence Zone (ITCZ). The MH-PI differences in temperature and precipitation give rise to important changes in isotope signal on a
global scale. More enriched $\delta^{18}O_p$ is simulated over the Arctic Ocean and Greenland due to summer warming, as well as over the western tropical Pacific led by decreased precipitation amount. Enhanced monsoonal rainfall over North Africa favors a pronounced decrease in isotope composition. Our simulated MH-PI anomalies in $\delta^{18}O_p$ are fairly consistent with speleothem and ice core reconstructions, though model-data mismatch can be found in certain regions (e.g., part of South America and East Antarctica). For the subtropical areas, isotopic compositions archived in speleothems might reflect very localized processes
related to the cave environment which are hardly to be captured by model simulations.

Analysis of the simulated isotope-temperature relationship reveals a stable spatial $\delta^{18}O_p$-T gradient across the MH and PI. On a global scale, the modeled spatial gradient (0.71 ‰/°C for both PI and MH) is comparable to the observed value (0.69 ‰/°C) (Dansgaard, 1964). This coefficient, however, may vary from region to region in response to a number of meteorological factors. Therefore, for meaningful reconstructions of past climate changes of a given area, attempts should be made to validate





the local $\delta^{18}O_p$-temperature gradient at the time of interest (Fricke and O'Neil, 1999). In the present study we calculate the MH-to-PI temporal gradient between $\delta^{18}O_p$ and temperature and find that the gradient over Greenland and Antarctica can become reasonable only if the variables are derived from the warmest month of the year. This result is consistent with the findings of Cauquoin et al. (2019). Furthermore, a clear amount effect is reflected in the temporal $\delta^{18}O_p$-precipitation relationship over the North Africa monsoon region, as a pronounced depletion in $\delta^{18}O_p$ is found in response to increased monsoonal rainfall.

Numerous studies have investigated the onset of the modern summer monsoon over West Africa, but for the mid-Holocene corresponding studies are lacking as the temporal resolutions of most proxy records are insufficient for such an analysis. Paleoclimate model simulations may provide an alternative approach to investigate the onset of past monsoon events. In this work, we evaluate the onset of the MH WASM based on both simulated precipitation and isotopic variables. To our knowledge, this is the first time that the onset of mid-Holocene WASM is examined by such an approach. We propose a new method for

defining the WASM onset using daily model values of $\delta^{18}O_p$, $dex_p$, and precipitation rate. Our simulations show that there has been no obvious difference between the MH and PI WASM onset, indicating that the simulated intensification in MH WASM is more likely attributable to a strengthening in rainfall rate than an extension of the monsoon duration.

*Code and data availability.*    The model source codes and raw model outputs related to the present study as well as a detailed instruction on how to compile AWIESM-wiso and perform PI and MH simulations are available from https://doi.org/10.5281/zenodo.7920091. A licence

agreement is required for using the atmosphere component (ECHAM6). The isotope component can be used upon request to the corresponding author MW (Martin.Werner@awi.de). The marine calcite data used in our study can be downloaded at https://doi.org/10.1594/PANGAEA.908831. The assimilated $\delta^{18}O$ data can be accessed from https://doi.pangaea.de/10.1594/PANGAEA.889922.

*Author contributions.*    MW and GL developed the original idea for this study. MW and AC implemented stable water isotopes into ECHAM6. XS and YS enhanced FESOM2 with 3 passive isotope tracers. XS coupled the isotope fluxes and concentrations between ECHAM6 and

FESOM2. LJ provided foraminifera records and gave very helpful suggestions and discussions. AC downloaded and extracted the ice core and speleothem data. QW and HY provided technique support for the original AWI-ESM and contributed to the code modification. MW implemented sea ice fractionation processes and further improved the coupling strategy. XS performed model simulations and data analysis under the supervison of MW and GL. All authors contributed to the discussion and paper writing.

*Competing interests.*    At least one of the (co-)authors is a member of the editorial board of Geoscientific Model Development.

*Acknowledgements.*    The present study is supported by German Federal Ministry of Education and Science (BMBF) PalMod II WP 3.3 (grant no. 01LP1924B) and the National Natural Science Foundation of China (NSFC) (grant no. 42206256). The simulations were conducted on Deutsche Klimarechenzentrum (DKRZ) and AWI supercomputer (Ollie). The authors declare that no competing interests exist.





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
