# Peer review of "Simulated stable water isotopes during the mid-Holocene and pre-industrial using AWI-ESM-2.1-wiso"

_Geoscientific Model Development, 2023_

## Author Response (AR1)

**Response letter**

Xiaoxu Shi1,2, Alexandre Cauquoin3, Gerrit Lohmann1,4, Lukas Jonkers4, Qiang Wang1, Hu Yang1,4, Yuchen Sun1,5, and Martin Werner1

1Alfred Wegener Institute, Helmholtz Center for Polar and Marine Research, Bremerhaven, Germany
 2Southern Marine Science and Engineering Guangdong Laboratory (Zhuhai), Zhuhai, China
 3Institute of Industrial Science, The University of Tokyo, Kashiwa, Japan
 4MARUM Center for Marine Environmental Sciences, University of Bremen, Bremen, Germany
 5Institute of Tibetan Plateau Research, Chinese Academy of Sciences, Beijing, China

**Dear Reviewers**,**

Thank you very much for your positive and constructive comments. In the following, we present our point-to-point responses. Our answers to your comments are written in blue.

Thanks again for your time and efforts.

5 Best,

Xiaoxu

**1 Comments from Reviewer 1**

Summary of Shi et al.

Shi et al. has performed a mid-Holocene (MH) time slice experiment with the new isotope enabled Earth System Model
AWI-ESM-wiso. The authors validate the control run and go on to compare the model with isotopic archives. Although the model performs well for the control run and the spatial pattern of the MH anomalies are correct the overall amplitude of the isotope anomalies appear underestimated, which could mean that the MH climate anomalies are underestimated. The authors investigate the spatial and temporal relations of isotopes and climate. One conclusion is that the modern spatial slope can be used as surrogate for the temporal slope in Greenland and Antarctica. Finally, the authors offer an alternative method employing
a combination of isotope anomalies and precipitation amount to determine the onset of the West African summer monsoon.

General comments.

I find the manuscript overall well-written, well-structured and with appropriate figures to illustrate the results. As the detailed comments below show I have a number of suggestions and concerns that I think should be taken into account before considering publication. One major concern is the somewhat superficial treatment of the debate on temporal versus spatial slope, which

20 is a topic that has been extensively researched, and I find some key references missing in this context. Temporal and spatial changes in climate are different in fundamental ways, and this is reflected in the difference in the temporal and spatial slope.

I think we are past the point where we are asking if the temporal and spatial slopes are interchangeable, which is summarized by the publications listed below (see comment to L304-307 and L319-324).

In summary I find that minor revisions should be made to accommodate the major and minor comments in this review.

**25 We really appreciate your positive comments and suggestions, which have significantly improved our manuscript. We have modified our paper accordingly. Details are as follows:**

Detailed comments.

Abstract

1. L2 "Straightforward" do you mean "direct"?

30 Thanks for the comment, we now changed "straightforward" to "direct"

2. L6 Move "well" to right before the comma.

We have modified the texts according to the comment.

3. L9-10 "The ratio of the MH-PI difference ... is reasonable ... " clarify if you are comparing to measured d180.

Thanks for the comment, we now changed the text into (see also L10 in the difference-tracked version):

35 *"The ratio of the MH-PI difference in*  $\delta^{18}O_p$  *to the MH-PI difference in surface air temperature is comparable to proxy records over Greenland and Antarctica only when summertime air temperature is considered."*

4. L39-43 The isotope-temperature relationship can also be affected by the proportion of continental sources and recycling of vapor (Werner et al. (2001), Sjolte et al., (2014)).

Thanks for the comment, we now changed the text into (L48-52 in the difference-tracked version):

- 40 "Changes in the primary moisture source areas or air mass transport trajectories during the LGM can have a significant impact on the isotope-temperature relationship, which has a potential to invalidate the isotopic paleothermometer approach based on the use of modern observations (Delaygue et al., 2000; Werner et al., 2001). Moreover, the isotope-temperature relationship can also be influenced by the proportion of continental sources and recycling of vapor under interglacial boundary conditions (Sjolte et al., 2014)."
- 45 5. L59 "Straightforward" do you mean "direct"?

Thanks for the comment, we now changed "straightforward" to "direct"

6. L139 Is spring equinox fixed to March 21? This information can be added to Table 1.

We now added this information into Table 1 (Page 6 in the difference-tracked version).

7. L172 "Straightforward" do you mean "direct"?

**50 In the revised version, we changed "straightforward" to "direct"**

8. Section 2.3.5 "Assimilation product". I suggest a more informative title such as "Marine d18O reanalysis data".

We have modified the texts according to the comment (L210 in the difference-tracked version). Moreover, the same change was also made for several other parts of the manuscript where "assimilation product" was used.

9. L214-215 High bias in d18O over Antarctica has also been linked to a low bias in water vapor for cold regions leading to
a precipitation weighting towards higher d18O (Masson-Delmotte et al., 2008).

Thanks for the comment, we now changed the text into (L234-235 in the difference-tracked version) "According to Masson-Delmotte et al. (2008), this underestimation can also be related to a low bias in water vapor for cold regions that favors higher  $\delta^{18}O_p$  values."

10. Figure 3/4/5/6. Are the simulated MH-PI anomalies significant?

60 In the updated manuscript, we performed Student's t-est to test the significance level of the anomalies shown in Fig. 3/4/5/6, and we marked the area with significance level of greater than 95% by black dots. We refer to Fig. 3/4/5/6 in our revised manuscript and the difference-tracked version.

11. Figure 6b Why not plot the RMSE between the different species which you write in the text? I think the reader expects this since it is included in the other plots. It doesn't take much to explain the details in the figure caption.

65 Thanks, we now added the RMSE across all species in Fig. 6b, and we also display the RMSE value for each individual foraminifera type in the corresponding legend. We refer to Fig. 6b in our revised manuscript and the difference-tracked version.

12. L304-307 I think it would be good to move this to the introduction and include some relevant paper discussing this (Cuffey et al., 1992; Sime et al., 2009; Sjolte et al., 2011; Kindler et al., 2014; Sjolte et al., 2014; Guan et al., 2016).

**70 Yes, we now added in the introduction (L38-46 in the difference-tracked version):**

"The observed present-day spatial slope between isotope and temperature is widely used as a surrogate for the temporal gradient at reconstruction sites (Cuffey et al., 1992; Sime et al., 2009; Kindler et al., 2014). A model simulation using isoCAM3 suggests that both the temporal slope and spatial slope remain largely stable throughout the last deglaciation (Guan et al., 2016). However, a significant region-dependency is found for both the temporal slope (Guan et al., 2016)

- 75 and the spatial slope (Sjolte et al., 2011). Guan et al. (2016) also point out that the temporal slope is usually smaller than the spatial slope in the extratropics. Besides, some studies indicate that the temporal relationship between isotopes and temperature may vary over time and the temporal isotope-temperature slope may differ from the observed spatial slope at present-day (e.g., Werner et al., 2000; Sjolte et al., 2014). Therefore, using isotope data from different archives to draw quantitative inferences about past climate variability remains challenging."
- 13. Figure 8: It is risky to calculate a slope between only two points, but one could use the STD of the interannual variability to estimate the uncertainty or use the annual/seasonal data to make bootstrap estimates of the uncertainty in slope.

In the revised manuscript, we have more analysis on the temporal slopes, we calculated the slopes not only for annual mean temperature, but also for JJAS and DJFM seasons. For more details please refer to our responses to your comments number 16 and 18.

85 14. L323 The estimate is affected by the selection of the data. For example, if coastal data is included or only higher altitude sites on the ice sheet (Siolte et al., 2011).

Thanks for the comment, we now changed the text into (L359-360 in the difference-tracked version): "In addition, the spatial  $\delta^{18}O_p$ -temperature relationship can be affected by the criteria of data selection, e.g., if coastal areas or only higher altitude sites on the ice sheet are included (Siolte et al., 2011)."

- 90 15. L319-324 Vinther et al. (2009) estimated a slope of 0.5 permil/degree C using borehole temperature and ice core data. I think it is well documented from the papers I mention above that the spatial slope and temporal slope are not interchangeable and that there are large regional differences in both. To quote Guan et al. (2016): "Finally, the relation between temporal and spatial slopes is understood using a semiempirical equation that is derived based on both the Rayleigh distillation and a fixed spatial slope. The slope equation quantifies the Boyle's mechanism and suggests that the temporal slope is usually smaller than
- the spatial slope in the extratropics mainly because of the polar amplification feature in global climate change, such that the 95 response in local temperature at middle and high latitudes is usually greater than that in the total equivalent source temperature."

Thanks, we added this estimation of greenland spatial slope in our text (see also L345-351 in the difference-tracked version):

"Over Greenland, our modeled  $\delta^{18}O_{n}$ -temperature gradients under present-day and MH conditions are 0.76 $\pm$ 0.042 and  $0.74\pm0.045$  % of C, respectively (Fig.7b,g), higher than the value obtained from modern observations (0.67 % of C) (Johnsen 100 et al., 1989) and previous model studies using MPI-ESM-wiso (0.71 %/°C) (Cauquoin et al., 2019) and ECHAM4 (0.58  $\% d^{\circ} C$ ) (Werner et al., 2000). However, a much smaller Holocene spatial isotope-temperature slope, ranging from 0.43  $\% d^{\circ} C$ to 0.53 % J°C, was estimated based on ice core records and borehole temperatures for the Greenland ice sheet (Vinther et al., 2009). "

Besides, in the end of section 5.1, as also added (see L361-364 in the difference-tracked version): 105

"It is important to note that there is a significant region-dependency found for the spatial isotope-temperature slope (Sjolte et al., 2011), and that the spatial slope might differ from the temporal slope due to changes in the seasonality of precipitation (e.g., Werner et al., 2000), moisture source regions (e.g., Delaygue et al., 2000) or polar amplification feature (Guan et al., 2016). In the following section, we aim to analyze the temporal relationship between our simulated  $\delta^{18}O_p$  and climate variables."

16. L341 What is the choice of 0.5 degree C anomalies based on? Can't you base it on whether the anomalies are significant?

Regarding the first question, for numerical reasons, a threshold of 0.5 C (or 1 C) is chosen here to avoid the situation that a small temperature anomaly causes an enormous unrealistic isotope-temperature gradient. On the other hand, the

110

threshold can guarantee that the isotope anomaly is mostly affected by temperature changes rather than other elements

- 115 such as precipitation. We now clarified this point in the revised paper (see L371-374 in the difference-tracked version): 115 "To avoid numerical errors in calculated temporal relationships caused by very small MH-PI temperature changes, another criterion is adopted that the absolute change of T between the two time intervals (i.e.,  $T_{MH} - T_{PI}$ ) must be nonnegligible, namely not less than 0.5 °C. Applying such a threshold can also ensure a temperature-dependency of the isotope changes''
- 120 Regarding the second question, it is a good idea to calculate the isotope-temperature gradient based on the significance of the temperature anomalies. The results are shown here in Fig. R1 in this response letter. From Fig. R1a we see that the data on Greenland is missing, this is because of the minor change in annual mean temperature between MH and PI, which is statistically not significant for those regions (This can also be seen from the temperature anomaly plot, i.e., Fig. 3c of the paper). But the value for the West Antarctica (0.61 %d°C) are more close to observation compared to other T definitions. If we only consider the warmest month, as seen in Fig. R1b, the pattern is almost the same as in Fig. 8c of the paper, and the detailed numbers for the isotope-temperature gradients are also very similar (Greenland: 0.64

 $% d^{\circ}C$ ; Antarctic: 0.48  $% d^{\circ}C$ ; East Antarctica: 0.55  $% d^{\circ}C$ ; West Antarctica: 0.39  $% d^{\circ}C$ ).

135

17. L363 Judging by Figure 8d there is still quite some regional differences in slope. What is the slope of key ice core locations in Greenland and Antarctica, and what is the reconstructed temperature anomalies from ice core d18O based in this130 slope? This can then be compared to the simulated temperature anomaly.

Yes there are regional difference and it is a good idea to compared modeled temperature anomalies with ice core records. In addition, the slope of key ice core locations in Greenland and Antarctica has been illustrated in Table 2 of the manuscript, and we can use these slopes, along with isotope anomalies documented by ice cores, to compute the temperature anomalies for each ice core location. Please see Fig. R2 for detailed information, this figure is also provided in the revised paper (see Fig. 9 of the paper). We also add a table in the paper to illustrate observed temperature anomalies and modeled T anomalies for each region and for each T definition (please refer to Table 3 in the updated paper). In addition, we added the following texts (see also 420-428 in the difference-tracked version):

"For each defined temperature (T), we calculate the average temperature anomalies (MH-minus-PI T anomalies) and compare the results with temperature reconstructions based on ice core  $\delta^{18}O_p$  values and observed modern spatial  $\delta^{18}O_p$

- 140 -temperature slopes. As shown in Fig. 9a, there is a lack of consensus between the simulated MAT anomalies and the values obtained from proxy records in Greenland. However, it is evident that the MTWA and T-monsoon exhibit a relatively stronger agreement with the observed data (Fig. 9b,c). Table 3 indicates that the proxy-based mean temperature anomaly over Greenland is 1.01 °C. Notably, the MTWA0.5 and sig. MTWA definitions provide a mean temperature anomaly of 1.29 °C across the Greenland ice core locations, which is relatively closer to the observed data compared to other T definitions.
- 145 In Antarctica, non-uniform temperature anomalies are observed, with modest changes in some areas and more significant cooling and warming in other regions. However, our model shows a general cooling trend across the entire Antarctic continent for all T definitions (Fig. 9d-f)."